# Semantic Parsing with Candidate Expressions for Knowledge Base Question Answering

## Abstract

Semantic parsers convert natural language to logical forms, which can then be evaluated on knowledge bases (KBs) to produce denotations. Recent semantic parsers have been developed with sequence-to-sequence (seq2seq) pre-trained language models (PLMs) or large language model, where the models treat logical forms as sequences of tokens. For syntactic and semantic validity, the semantic parsers use grammars that enable constrained decoding. However, the grammars lack the ability to utilize large information of KBs, although logical forms contain representations of KB components, such as entities or relations. In this work, we propose a grammar augmented with *candidate expressions* for semantic parsing on large KBs with a seq2seq PLM.[1] The grammar defines actions as production rules, and our semantic parser predicts actions during inference under the constraints by types and candidate expressions. We apply the grammar to knowledge base question answering, where the constraints by candidate expressions assist a semantic parser to generate valid KB components. Experiments on the KQAPRO benchmark showed that the constraints by candidate expressions increased the accuracy of our semantic parser, and our semantic parser achieved state-of-the-art performance on KQAPRO.

## 1 Introduction

Semantic parsing is the task of mapping natural language to logical forms, which can be evaluated on given knowledge bases (KBs) to produce corresponding denotations. For example, a question answering system can use a semantic parser to convert a user's question to a query (logical form), then the query derives an answer (denotation) from a database (KB) (Zelle & Mooney, 1996; Cai & Yates, 2013). Traditional semantic parsers depend on grammars with lexicons that map spans of utterances to atomic units, which are subsequently composed into logical forms by following the grammars (Zettlemoyer & Collins, 2005; Wong & Mooney, 2007; Liang et al., 2011). In contrast, the emergence of sequence-to-sequence (seq2seq) frameworks (Sutskever et al., 2014; Bahdanau et al., 2015) have led to the development of neural semantic parsers whose neural networks convert natural language token sequences to action sequences that construct logical forms (Jia & Liang, 2016; Dong & Lapata, 2016).

Neural semantic parsers have used grammars that utilize types for constrained action decoding, in which the actions are designed to generate only well-typed logical forms. The actions can be defined as production rules that expand typed placeholders into sub-expressions of logical forms (Yin & Neubig, 2017; Rabinovich et al., 2017; Krishnamurthy et al., 2017), or as typed atomic units that are inserted into partially constructed logical forms (Guu et al., 2017; Cheng et al., 2017; Liang et al., 2017; Dong & Lapata, 2018; Goldman et al., 2018). In particular, semantic parsers that take production rules as actions are easily adapted to diverse applications with different logical forms, once the corresponding production rules are defined.

Recent work has incorporated grammars into semantic parsers based on seq2seq pre-trained language models (PLMs) (Lewis et al., 2020; Raffel et al., 2020), or based on large language models (LLMs) (Brown et al., 2020; Chen et al., 2021a), where the models have specific decoders with tokenizers. The semantic parsers sequentially generate tokens that extend prefixes of logical forms

---

[1]Our code will be publicly available if this paper is accepted.

Figure 1: Semantic parsing on an example of KQAPRO. $k$ is a KB, $x$ is an utterance, $\boldsymbol{a}$ is an action sequence, $r(\boldsymbol{a})$ is an intermediate representation built by $\boldsymbol{a}$, $l(r(\boldsymbol{a}))$ is a logical form which corresponds to $r(\boldsymbol{a})$, and $[\![l(r(\boldsymbol{a}))]\!]_k$ is the denotation when $l(r(\boldsymbol{a}))$ is evaluated on $k$.

under the guidance of the grammars that keep the prefixes always valid. The semantic parsers use context-free grammars (CFGs) for syntactic validity (Wu et al., 2021; Shin et al., 2021; Wang et al., 2023), and additionally uses context-sensitive constraints for semantic validity (Scholak et al., 2021; Poesia et al., 2022).

However, the grammars for semantic parsing with specific decoders lack the ability to utilize *large* information of KBs. The information includes KB components, such as entities or relations, and categories that the components belong to. Since logical forms contain representations of KB components, the information of KBs is necessary for semantic parsers to generate valid logical forms. Therefore, incorporating the large information of KBs into grammars is important, but scalable and efficient designs of the grammars are inevitable for practical use.

In this work, we propose a grammar augmented with *candidate expressions* for semantic parsing on large KBs with a seq2seq PLM. Our grammar combine previous constrained decoding methods that construct compositional structures (Yin & Neubig, 2017) and generate KB components (Cao et al., 2021a), which correspond to candidate expressions. The two different methods are seamlessly unified into our grammar which formulate constrained decoding as the problem of restricting actions for a given intermediate representation. In addition, we efficiently implemented the constrained decoding method with our grammar, then the method has small overhead during decoding.

We experiment on KQAPRO (Cao et al., 2022), which is a benchmark for large-scale complex knowledge base question answering (KBQA). Our semantic parser is based on BART (Lewis et al., 2020), and the semantic parser is fine-tuned with supervision on action sequences. Experimental results show that the constraints by candidate expressions increase accuracy of our semantic parser. Our semantic parser with the proposed grammar achieved state-of-the-art performance on KQAPRO.

## 2 SEMANTIC PARSING

We first formally define a semantic parser as a function $f_\theta : \mathcal{X} \to \mathcal{A}$ that maps a natural language utterance $x \in \mathcal{X}$ to an action sequence $\boldsymbol{a} \in \mathcal{A} = \bigcup_{i \in \mathbb{N}} \mathcal{A}^i$ that builds an intermediate representation $r(\boldsymbol{a})$ which corresponds to a logical form $l(r(\boldsymbol{a}))$, which is evaluated on a KB $k$ to produce a denotation $[\![l(r(\boldsymbol{a}))]\!]_k$ (Figure 1). As a seq2seq model with a probability distribution over actions at each time step, we formulate a semantic parser as:

$$f_\theta(x) = \arg\max_{\boldsymbol{a} \in \mathcal{A}} p_\theta(\boldsymbol{a} \mid x) = \arg\max_{\boldsymbol{a} \in \mathcal{A}} \prod_{i=1}^{|\boldsymbol{a}|} p_\theta(a_i \mid \boldsymbol{a}_{1:i-1}, x) \tag{1}$$

where $\theta$ is the set of parameters of the model. In practice, a semantic parser finds a sub-optimal solution by greedy search or beam search within a limited number of operations.

Table 1: Subset of a grammar definition. Each row specifies the properties of a node class. About logical form templates, `@i` means the logical form of a child node for an index `i`, `@*` means the logical forms of all child nodes, and `#(expr)` means that the logical form is the result of the evaluation of `(expr)`.

| Class name | Return type | Parameter types | Logical form template |
|---|---|---|---|
| `program` | `result` | `[result]` | `@0` |
| `query-rel-qualifier` | `result-rel-q-value` | `[kw-relation kw-qualifier obj-entity obj-entity]` | `(query-rel-qualifier @2 @3 @0 @1)` |
| `keyword-relation` | `kw-relation` | `[kp-relation &rest kp-relation]` | `#(concat @*)` |
| `(nlt ·country)` | `(ut kp-entity kp-relation ...)` | N/A | `·country` |

Table 2: Example of building an intermediate representation by taking actions. For each step, an action expands the leftmost non-terminal, written in bold, into a logical form expression, underlined in the next step.

| Step | Intermediate representation |
|---|---|
| 0 | `(program `**`<result>`**`)` |
| 1 | `(program ``(query-rel-qualifier `**`<kw-relation>`**` <kw-qualifier> <obj-entity> <obj-entity>)``)` |
| 2 | `(program (query-rel-qualifier ``(keyword-relation `**`<kp-relation>`**` <kp-relation>*)`` <kw-qualifier> <obj-entity> <obj-entity>))` |
| 3 | `(program (query-rel-qualifier (keyword-relation ``(nlt ·country)`` `**`<kp-relation>`**`*) <kw-qualifier> <obj-entity> <obj-entity>))` |
| 4 | `(program (query-rel-qualifier (keyword-relation (nlt ·country) ``(nlt ·citizenship)`` `**`<kp-relation>`**`*) <kw-qualifier> <obj-entity> <obj-entity>))` |
| 5 | `(program (query-rel-qualifier (keyword-relation (nlt ·country) (nlt ·citizenship) ``reduce``) `**`<kw-qualifier>`**` <obj-entity> <obj-entity>))` |

The semantic parser learns to predict an action sequence $\boldsymbol{a}$ when given an utterance $x$ by maximizing the objective $J(D, \theta)$ with respect to parameters $\theta$ for a training set $D$:

$$J(D, \theta) = \sum_{(x, \boldsymbol{a}) \in D} \log p_\theta(\boldsymbol{a} \mid x). \tag{2}$$

However, to use a seq2seq PLM, we should reduce the discrepancy in formats between the actions and the natural language tokens that are predicted by the seq2seq PLM.

To adapt a seq2seq PLM to our semantic parsing framework, we divide $\mathcal{A}$, which is the set of actions, into two subsets: $\mathcal{A}^{\text{COM}}$ which contains actions that build **com**positional structures or atomic units, and $\mathcal{A}^{\text{NLT}}$ which contains actions that generate **n**atural **l**anguage **t**okens (Yin & Neubig, 2017). An action in $\mathcal{A}^{\text{NLT}}$ constructs a node `(nlt *)` where `*` is a natural language token. Then, (1) the embedding of an action in $\mathcal{A}^{\text{COM}}$ is learned from scratch and (2) the embedding of an action in $\mathcal{A}^{\text{NLT}}$ is fine-tuned from the pre-trained embedding of the corresponding token.

## 3 GRAMMARS WITH TYPES

An action $a$ is a production rule that is applied to the leftmost non-terminal $\nu(r(\boldsymbol{a}^\star))$ in an intermediate representation $r(\boldsymbol{a}^\star)$ built from a past action sequence $\boldsymbol{a}^\star$. The action $a = \alpha(c)$ corresponds to a node class $c$ that is defined by a grammar that specifies a return type and parameter types for $c$ (Table 1). The action $a$ expands the return type of $c$ to an expression that is composed of the name of $c$ and the parameter types of $c$:

$$\texttt{<return-type>} \rightarrow (\texttt{class-name <param-type-0> <param-type-1> ...})$$

or $a$ expands to the name of $c$ when $c$ does not have any parameter type:

$$\texttt{<return-type>} \rightarrow \texttt{class-name}$$

where the types become non-terminals. We express an intermediate representation as an S-expression which consists of symbols and parentheses (McCarthy, 1978). The S-expression is a tree structure in which the first symbol in a pair of parentheses is the parent node and the remaining symbols or sub-expressions are child nodes.

Under type constraints, an action $a$ can be applied to the leftmost non-terminal $\nu(r(\boldsymbol{a}^\star))$ when $a$'s left-hand side, $\kappa(a)$, and $\nu(r(\boldsymbol{a}^\star))$ have the same type or compatible types (Yin & Neubig, 2017; Krishnamurthy et al., 2017) (Figure 2). We define three special conditions for type compatibility:

**Sub-type inference** allows an action $a$ to be applied to the leftmost non-terminal $\nu(r(\boldsymbol{a}^\star))$ when the left-hand side $\kappa(a)$ has a sub-type of $\nu(r(\boldsymbol{a}^\star))$. For example, $a = \alpha(\texttt{query-rel-qualifier})$ has $\texttt{<result-rel-q-value>}$ as $\kappa(a)$, then $a$ can be applied to $\nu(r(\boldsymbol{a}^\star)) = \texttt{<result>}$, as $\texttt{result-rel-q-value}$ is a sub-type of $\texttt{result}$.

**Union types** allow the left-hand side $\kappa(a)$ of an action $a$ to have multiple types, then $a$ can be applied to the leftmost non-terminal $\nu(r(\boldsymbol{a}^\star))$ when the type of $\nu(r(\boldsymbol{a}^\star))$ is same or compatible with a type that belongs to $\kappa(a)$. We assign a union type to $\kappa(a)$ for an action $a = \alpha(\texttt{(nlt *)})$. For example, $a = \alpha\big(\texttt{(nlt ·country)}\big)$ has $\texttt{<kp-entry kp-relation ...>}$ as $\kappa(a)$, whose type is $\texttt{(ut kp-entry kp-relation ...)}$, then $a$ can be applied to $\nu(r(\boldsymbol{a}^\star))$ such as $\texttt{<kp-entry>}$ or $\texttt{<kp-relation>}$, but it cannot be applied to $\texttt{<vp-quantity>}$ which requires another action $a'$, such as $\alpha\big(\texttt{(nlt ·7)}\big)$, whose left-hand side $\kappa(a')$ is $\texttt{<vp-quantity ...>}$.

**Repeated types** allow the leftmost non-terminal $\nu(r(\boldsymbol{a}^\star))$ that has $*$ as a suffix to be repeated until a special action $* \rightarrow \texttt{reduce}$ is taken (Yin & Neubig, 2017). The special non-terminal that has $*$ is derived from a parameter type declared with the $\texttt{\&rest}$ keyword. For example, a node class $\texttt{keyword-relation}$ has $\texttt{kp-relation}$ as the second parameter type, which is declared with the $\texttt{\&rest}$ keyword, then the type becomes a non-terminal $\texttt{<kp-relation>}*$, which is repeated as $\nu(r(\boldsymbol{a}^\star))$ until $* \rightarrow \texttt{reduce}$ is taken.

The parsing procedure is to sequentially take actions, which expand the leftmost non-terminals to sub-expressions, until no non-terminal exists (Table 2). When the past action sequence is $\boldsymbol{a}_{1:t-1}$, an action $a_t$ replaces the leftmost non-terminal $\nu(r(\boldsymbol{a}_{1:t-1}))$ with the right-hand side of $a_t$, then the intermediate representation is updated as $r(\boldsymbol{a}_{1:t})$. For each step during parsing, a semantic parser should distinguish which actions are valid in the current intermediate representation. Therefore, a semantic parser needs a function $\Psi : \mathcal{R} \rightarrow 2^{\mathcal{A}}$ that maps an intermediate representation $r(\boldsymbol{a}^\star) \in \mathcal{R}$ to a set of valid actions $\Psi(r(\boldsymbol{a}^\star)) \subset \mathcal{A}$.

We define $\Psi^{\text{TYPE}}(r(\boldsymbol{a}^\star))$ as the set of all valid actions with respect to types. For an action $a \in \Psi^{\text{TYPE}}(r(\boldsymbol{a}^\star))$, the leftmost non-terminal $\nu(r(\boldsymbol{a}^\star))$ and the left-hand side $\kappa(a)$ have compatible types. Therefore, $\Psi^{\text{TYPE}}$ guides a semantic parser to gradually compose well-typed intermediate representations. When parsing is finished, the final expression is a complete intermediate representation $r(\boldsymbol{a})$, which can be converted to a logical form $l(r(\boldsymbol{a}))$, as each node has a corresponding logical form template (Table 1).

# 4 CANDIDATE EXPRESSIONS

Our grammar with types build compositional structures of intermediate representations, but the grammar is insufficient to synthesize valid KB components. A KB component is constructed by a node, such as $\texttt{keyword-relation}$, and the node has a sequence of $\texttt{(nlt *)}$ nodes as children. Unless the sequence of $\texttt{(nlt *)}$ nodes becomes an existing KB component, a logical form that involves the sequence cannot produce a meaningful denotation.

We augment the grammar with candidate expressions to generate existing KB components. A candidate expression $e \in \mathcal{E}(c)$ for a node class $c$ is a predefined instance of a specific KB component category that corresponds to $c$. For example, the KB component category "relation", which corresponds to a node class $c = \texttt{keyword-relation}$, has predefined instances, such as $\texttt{"country of citizenship"}$ and $\texttt{"country for sport"}$, as candidate expressions $\mathcal{E}(c)$ (Table 3). The candidate expressions $\mathcal{E}(c)$ are shared with a node $o$ that is instantiated from the node class $c$; therefore $\mathcal{E}(c) = \mathcal{E}(o)$.

Table 3: Node classes, subsets of their candidate expressions, and the total numbers of candidate expressions.

| Class name | Subset of candidate expressions | Quantity |
|---|---|---|
| keyword-concept | "human", "music", "chief executive officer", "Academy Awards" | 791 |
| keyword-entity | "United States of America", "Nobel Peace Prize", "Cary Grant" | 14,471 |
| keyword-relation | "affiliation", "country of citizenship", "country for sport" | 363 |
| keyword-attribute-string | "DOI", "ISSN", "catalog code", "media type", "GitHub username" | 403 |
| keyword-attribute-number | "height", "width", "speed", "price", "radius", "melting point" | 201 |
| keyword-attribute-time | "date of birth", "work period (start)", "production date" | 25 |
| keyword-qualifier-string | "place of publication", "appointed by", "official website" | 226 |
| keyword-qualifier-number | "proportion", "ranking", "frequency", "number of subscribers" | 34 |
| keyword-qualifier-time | "start time", "end time", "point in time", "last update" | 15 |
| constant-unit | "mile", "inch", "gram", "hour", "year", "square kilometre" | 118 |

```
r(a⋆):  (program (query-rel-qualifier (keyword-relation (nlt ·country) <kp-relation>*)
                                       <kw-qualifier> <obj-entity> <obj-entity>))

Ψ^TYPE(r(a⋆)):  {<kp-relation ...> → (nlt ·of), <kp-relation ...> → (nlt ·with), ...}

Ψ^CAND(r(a⋆)):  {<kp-relation ...> → (nlt ·of), <kp-relation ...> → (nlt ·for), ...}
```

Figure 2: Example of two action sets $\Psi^{\text{TYPE}}(r(\boldsymbol{a}^\star))$ and $\Psi^{\text{CAND}}(r(\boldsymbol{a}^\star))$ for an intermediate representation $r(\boldsymbol{a}^\star)$. `<kp-relation>*` is $\nu(r(\boldsymbol{a}^\star))$ which is the leftmost non-terminal in $r(\boldsymbol{a}^\star)$, and `keyword-relation` is $\rho(\nu(r(\boldsymbol{a}^\star)))$ which is the parent node of $\nu(r(\boldsymbol{a}^\star))$. `<kp-relation ...> → (nlt ·of)` and `<kp-relation ...> → (nlt ·for)` are actions that result in valid prefixes of candidate expressions. `<kp-relation ...> → (nlt ·with)` is an action that results in an invalid prefix of a candidate expression. `<kp-relation ...>` is $\kappa(a)$ which is the left-hand side of $a \in$ `<kp-relation ...> → (nlt ·of) | (nlt ·with) | (nlt ·for)`.

We define $\Psi^{\text{CAND}}(r(\boldsymbol{a}^\star))$ as the set of valid actions with respect to candidate expressions. $\Psi^{\text{CAND}}(r(\boldsymbol{a}^\star))$ depends on $\rho(\nu(r(\boldsymbol{a}^\star)))$ which is the parent node of the leftmost non-terminal $\nu(r(\boldsymbol{a}^\star))$ (Figure 2). The parent node $\rho(\nu(r(\boldsymbol{a}^\star)))$ has (nlt *) nodes as children, whose concatenation should be always a prefix of a candidate expression $e \in \mathcal{E}(\rho(\nu(r(\boldsymbol{a}^\star))))$. Therefore, an action $a_t \in \Psi^{\text{CAND}}(r(\boldsymbol{a}_{1:t-1}))$ adds an (nlt *) node as a child to $\rho(\nu(r(\boldsymbol{a}_{1:t-1})))$, then the new concatenation of child nodes of $\rho(\nu(r(\boldsymbol{a}_{1:t})))$ becomes an extended prefix of a candidate expression $e \in \mathcal{E}(\rho(\nu(r(\boldsymbol{a}_{1:t-1}))))$.

We implement $\Psi^{\text{CAND}}$ with *trie* data structures (Cormen et al., 2009) that store candidate expressions which are split into natural language tokens (Cao et al., 2021a; Shu et al., 2022). For each node class $c$, we convert its candidate expressions $\mathcal{E}(c)$ into token sequences, which are then added to the trie $\tau(c)$. The trie $\tau(c)$ is shared with a node $o$ instantiated from the node class $c$; therefore $\tau(c) = \tau(o)$. A constructed trie $\tau(o)$ takes a token sequence as a prefix of a candidate expression $e \in \mathcal{E}(o)$, then retrieves valid tokens that can extend the prefix. Therefore, an action $a \in \Psi^{\text{CAND}}(r(\boldsymbol{a}^\star))$ is represented as $<...> \to$ (nlt *) where the token * is retrieved from the trie $\tau(\rho(\nu(r(\boldsymbol{a}^\star))))$ when given a token sequence from child nodes of $\rho(\nu(r(\boldsymbol{a}^\star)))$. Previous work has used one trie for entities (Cao et al., 2021a) or two distinct tries for predicates (Shu et al., 2022), whereas we use a distinct trie for each node class that corresponds to a KB component category.

Finally, we introduce $\Psi^{\text{HYBR}}$ which is a hybrid function of $\Psi^{\text{TYPE}}$ and $\Psi^{\text{CAND}}$. For an intermediate representation $r(\boldsymbol{a}^\star)$, $\Psi^{\text{HYBR}}$ returns a set of valid actions from $\Psi^{\text{CAND}}(r(\boldsymbol{a}^\star))$ when candidate expressions are defined for $\rho(\nu(r(\boldsymbol{a}^\star)))$, or from $\Psi^{\text{TYPE}}(r(\boldsymbol{a}^\star))$ otherwise:

$$\Psi^{\text{HYBR}}(r(\boldsymbol{a}^\star)) = \begin{cases} \Psi^{\text{CAND}}(r(\boldsymbol{a}^\star)) & \text{if HASCANDEXPR}(\rho(\nu(r(\boldsymbol{a}^\star)))) \\ \Psi^{\text{TYPE}}(r(\boldsymbol{a}^\star)) & \text{otherwise.} \end{cases} \tag{3}$$

Therefore, $\Psi^{\text{HYBR}}$ uses $\Psi^{\text{TYPE}}$ to construct compositional structures, and uses $\Psi^{\text{CAND}}$ to generate KB components, which are attached to the compositional structures.

## 5 IMPLEMENTATION DETAILS AND EXPERIMENTAL SETUP

We implement our semantic parser with the proposed grammar on KQAPRO, which is a large-scale benchmark for KBQA (Cao et al., 2022).

**Datasets.** We use the standard KQAPRO data splits: the training set $D^{\text{TRAIN}}$, the validation set $D^{\text{VAL}}$ and the test set $D^{\text{TEST}}$; they contain 94,376, 11,797 and 11,797 examples respectively (Cao et al., 2022). Each example includes a question, a logical form written in KoPL (Cao et al., 2022) and an answer. We map a question to an utterance $x$, and an answer to a gold denotation $y$. We also augment an example in $D^{\text{TRAIN}}$ with an action sequence $\boldsymbol{a}$, which is converted from the KoPL logical form of the example. The average, maximum and minimum length of $\boldsymbol{a}$ in $D^{\text{TRAIN}}$ is 28.8, 149 and 8 respectively.

**Models.** We develop our semantic parser with BART (Lewis et al., 2020), which is a seq2seq PLM. For a fair comparison with previous work, we especially use the BART-base model, with which previous semantic parsers are developed (Cao et al., 2022; Nie et al., 2022; 2023).

**Grammars.** Our grammar defines the actions in $\mathcal{A} = \mathcal{A}^{\text{COM}} \cup \mathcal{A}^{\text{NLT}}$ (Section 2), where $|\mathcal{A}^{\text{COM}}|$ is 53 and $|\mathcal{A}^{\text{NLT}}|$ is 50,260, which is same with the number of non-special tokens of BART. From the grammar, different $\Psi$ functions are derived: (1) $\Psi^{\text{HYBR}}$, (2) $\Psi^{\text{TYPE}}$, (3) $\Psi^{\text{TYPE}^-}$ which replaces different union types with the same type and (4) $\Psi^{\text{NONE}}$ which always returns $\mathcal{A}$, the set of all actions, without applying any constraint. The action sets that are returned from the four functions have the following subset relations:

$$\Psi^{\text{HYBR}}(r(\boldsymbol{a}^\star)) \subset \Psi^{\text{TYPE}}(r(\boldsymbol{a}^\star)) \subset \Psi^{\text{TYPE}^-}(r(\boldsymbol{a}^\star)) \subset \Psi^{\text{NONE}}(r(\boldsymbol{a}^\star)) = \mathcal{A}. \tag{4}$$

We address the effect of the functions $\boldsymbol{\Psi} = \{\Psi^{\text{HYBR}}, \Psi^{\text{TYPE}}, \Psi^{\text{TYPE}^-}, \Psi^{\text{NONE}}\}$ in Section 7.1.

**Intermediate representations.** An intermediate representation $r(\boldsymbol{a}^\star)$ is stored in a linked list that consists of nodes and complete sub-expressions. An action $a$ that is not $* \rightarrow \texttt{reduce}$ attaches a node to the linked list. When the parent node $\rho(\nu(r(\boldsymbol{a}^\star)))$ has no more non-terminal as a child node, or when the last action is $* \rightarrow \texttt{reduce}$, $\rho(\nu(r(\boldsymbol{a}^\star)))$ and its children are popped from the linked list, then they are again attached to the linked list as a complete sub-expression (Cheng et al., 2017). Since a linked list can be shared as a sub-linked list for other linked lists, search algorithms do not need to the copy the previous intermediate representation $r(\boldsymbol{a}_{1:t-1})$ when multiple branches with different actions from $\Psi(r(\boldsymbol{a}_{1:t-1}))$ occur for a time step $t$.

**Search.** Our semantic parser searches for an action sequence $\boldsymbol{a}$ when given an utterance $x$. A search algorithm, such as greedy search or beam search, depends on a scoring function:

$$s(a\,;\,\boldsymbol{a}_{1:t-1}, x, \theta) = \log p_\theta(a \mid \boldsymbol{a}_{1:t-1}, x) + \log p_\theta(\boldsymbol{a}_{1:t-1} \mid x) = \log p_\theta((\boldsymbol{a}_{1:t-1}, a) \mid x) \tag{5}$$

which assigns a priority to an action $a \in \Psi^{\text{NONE}}(\boldsymbol{a}_{1:t-1}) = \mathcal{A}$ as a candidate for the next action $a_t$. We replace the scoring function $s$ with $s_\Psi$ which uses $\Psi \in \{\Psi^{\text{HYBR}}, \Psi^{\text{TYPE}}, \Psi^{\text{TYPE}-}\}$:

$$s_\Psi(a\,;\,\boldsymbol{a}_{1:t-1}, x, \theta) = \begin{cases} s(a\,;\,\boldsymbol{a}_{1:t-1}, x, \theta) & \text{if } a \in \Psi(r(\boldsymbol{a}_{1:t-1})) \\ -\infty & \text{otherwise.} \end{cases} \tag{6}$$

We use greedy search and beam search in the *transformers* library (Wolf et al., 2020). The search implementation can take a scoring function $s_\Psi$ as an argument to predict $\boldsymbol{a}$ when given $x$. Our semantic parser uses greedy search by default and additionally uses beam search in Section 6.

We efficiently implement $s_\Psi$ and $\Psi^{\text{HYBR}}$, so the time cost for our method is small enough for practical applications (Appendix B). During evaluation on $D^{\text{VAL}}$ with batch size 64, the average time to predict $\boldsymbol{a}$ from $x$ by greedy search was (1) 3.8 milliseconds (ms) with $s$, and (2) 10.2 ms with $s_\Psi$ and $\Psi = \Psi^{\text{HYBR}}$ on our machine [2]; therefore, the time cost for $s_\Psi$ and $\Psi^{\text{HYBR}}$ was 6.4 ms. The time cost when using beam search with a beam size of 4 was 23.2 ms, as the cost is proportional to the beam size.

**Execution.** A search process finds an action sequence $\boldsymbol{a}$, from which an executable logical form $l(r(\boldsymbol{a}))$ is derived. The logical form $l(r(\boldsymbol{a}))$ is written as an S-expression, so a transpiler (Odendahl, 2019) converts $l(r(\boldsymbol{a}))$ into Python code on the fly, then the code is executed over a KB $k$ to produce the denotation $[\![l(r(\boldsymbol{a}))]\!]_k$.

---

[2]CPU = Ryzen Threadripper PRO 5975WX, GPU = NVIDIA GeForce RTX 3090

Table 4: Accuracies on the overall $D^{\text{VAL}}$, the overall $D^{\text{TEST}}$ and each category of examples in $D^{\text{TEST}}$.

| Model | $D^{\text{VAL}}$ | $D^{\text{TEST}}$ | | | | | | | |
|---|---|---|---|---|---|---|---|---|---|
| | Over-all | Over-all | Multi-hop | Quali-fier | Comp-arison | Logi-cal | Count | Verify | Zero-shot |
| BART KoPL (Cao et al., 2022) | – | 90.55 | 89.46 | 84.76 | 95.51 | 89.30 | 86.68 | 93.30 | 89.59 |
| GraphQ IR (Nie et al., 2022) | – | 91.70 | 90.38 | 84.90 | 97.15 | **92.64** | **89.39** | **94.20** | **94.20** |
| Semantic Anchors (Nie et al., 2023) | – | 91.72 | – | – | – | – | – | – | – |
| Ours with $\Psi^{\text{NONE}}$ or $\Psi^{\text{TYPE}-}$ | 92.08 | 91.74 | 90.74 | 86.88 | 97.05 | 90.38 | 87.06 | 93.51 | 90.55 |
| Ours with $\Psi^{\text{TYPE}}$ | 92.08 | 91.75 | 90.76 | 86.88 | 97.05 | 90.41 | 87.13 | 93.51 | 90.55 |
| Ours with $\Psi^{\text{HYBR}}$ | 92.96 | 92.60 | 91.51 | 87.73 | 97.47 | 91.37 | 87.96 | **94.20** | 91.31 |
| + beam size = 4 | **93.17** | **92.81** | **91.67** | **88.12** | **97.57** | 91.73 | 88.11 | 94.06 | 91.88 |

**Evaluation.** As an evaluation measure, we use denotation accuracy, which is the fraction of examples where the predicted denotation $[\![l(r(\boldsymbol{a}))]\!]_k$ and the gold denotation $y$ are identical.

**Training procedure.** Our semantic parser $f_\theta$ learns to predict an action sequence $\boldsymbol{a}$ from a given utterance $x$. At each epoch, we optimize the parameters $\theta$ by maximizing the objective $J(D^{\text{TRAIN}}, \theta)$ (Eq. 2), and evaluate $f_\theta$ with each $\Psi \in \boldsymbol{\Psi}$ on $D^{\text{VAL}}$. Once the training is complete, each $\Psi \in \boldsymbol{\Psi}$ has a checkpoint of parameters, with which $\Psi$ achieves the highest accuracy on $D^{\text{VAL}}$ during training. In Sections 6 and 7, we report accuracies by the checkpoints.

**Hyperparameters.** We adapt the hyperparameters for training from BART KoPL (Cao et al., 2022), which is a previous semantic parser on KQAPRO. The number of epochs is 25. The batch size is 16, which is much smaller than that of other previous work (Nie et al., 2022; 2023), whose batch size is 128. For each update on parameters when given a batch, the learning rate linearly increases from 0 to 3e-5 for the first 2.5 epochs, then linearly decreases to 0. The objective Eq. 2 is optimized by AdamW (Loshchilov & Hutter, 2017), which takes the learning rate and other arguments with the following values; $\beta_1$ is 0.9, $\beta_2$ is 0.999, $\epsilon$ is 1e-8 and the weight decay rate $\lambda$ is 1e-5.

## 6 MAIN RESULTS

We report the accuracies of our semantic parers, and compare the accuracies with those of previous semantic parsers (Cao et al., 2022; Nie et al., 2022; 2023) (Table 4). The accuracies are computed on the overall $D^{\text{VAL}}$, the overall $D^{\text{TEST}}$ and each category of examples in $D^{\text{TEST}}$ (Cao et al., 2022). Since the semantic parser with $\Psi^{\text{NONE}}$ achieved the same result as that of $\Psi^{\text{TYPE}-}$, we report their accuracies without duplication.

The previous semantic parsers are BART KoPL (Cao et al., 2022), GraphQ IR (Nie et al., 2022) and Semantic Anchors (Nie et al., 2023). The three previous semantic parsers, as well as ours, are developed with BART-base. The BART KoPL model predicts logical forms written in KoPL, which is linearized in postfix representations. The GraphQ IR model predicts intermediate representations written in the GraphQ IR language, which resembles English. The Semantic Anchors model predicts logical forms written in SPARQL, and the model learns from sub-tasks about semantic anchors.

All of our semantic parsers achieved higher accuracies on the overall $D^{\text{TEST}}$ than the previous semantic parsers (Table 4). The model with $\Psi^{\text{NONE}}$ achieved decent accuracies without using any constraint during parsing; this shows that a seq2seq PLM can be effectively fine-tuned to predict a sequence of actions that are production rules. The model with $\Psi^{\text{TYPE}}$ slightly increased our accuracies on $D^{\text{TEST}}$. The model with $\Psi^{\text{HYBR}}$ noticeably increased our accuracies on $D^{\text{VAL}}$ and $D^{\text{TEST}}$. Finally, when the beam size was 4, the model with $\Psi^{\text{HYBR}}$ achieved the highest accuracies on the overall $D^{\text{VAL}}$ and the overall $D^{\text{TEST}}$.

However, our semantic parsers achieved lower accuracies than GraphQ IR on the categories of *Logical*, *Count* and *Zero-shot* in $D^{\text{TEST}}$. Ours and GraphQ IR have different designs of actions for intermediate representations: production rules for S-expressions, and tokens for English-like expressions. Therefore, the two designs generalize differently on specific categories of examples.

Table 5: Accuracies on $D^{\text{VAL}}$ with different functions in $\boldsymbol{\Psi}$.

| $\Psi \in \boldsymbol{\Psi}$ | Constraints | | | Number of training examples (percentage) | | | | | | |
|---|---|---|---|---|---|---|---|---|---|---|
| | Cand. expr. | Union types | Types | 94 (0.1 %) | 283 (0.3 %) | 944 (1 %) | 2.83k (3 %) | 9.44k (10 %) | 28.3k (30 %) | 94.4k (100 %) |
| $\Psi^{\text{HYBR}}$ | ✓ | ✓ | ✓ | 35.36 | 57.74 | 74.00 | 82.14 | 87.62 | 90.74 | 92.96 |
| $\Psi^{\text{TYPE}}$ | ✗ | ✓ | ✓ | 31.36 | 52.33 | 70.48 | 79.05 | 85.52 | 89.37 | 92.08 |
| $\Psi^{\text{TYPE}-}$ | ✗ | ✗ | ✓ | 31.29 | 52.31 | 70.45 | 79.01 | 85.49 | 89.36 | 92.08 |
| $\Psi^{\text{NONE}}$ | ✗ | ✗ | ✗ | 28.44 | 50.78 | 70.20 | 78.98 | 85.48 | 89.36 | 92.08 |

## 7 ABLATION STUDY

We address the effect of constraints by each $\Psi \in \boldsymbol{\Psi}$ in Section 7.1 and the effect of candidate expressions in Section 7.2. Since the number of examples in $D^{\text{TRAIN}}$ is 94,376, which is a large number, the models that are trained on $D^{\text{TRAIN}}$ have high accuracies, whose difference is then small. Therefore, we train semantic parsers on various subsets of $D^{\text{TRAIN}}$, where the size of a subset changes exponentially and a larger subset includes a smaller subset. In addition, we evaluate the semantic parsers with $D^{\text{VAL}}$ instead of $D^{\text{TEST}}$, since the denotations in $D^{\text{TEST}}$ are not publicly available.

### 7.1 EFFECT OF CONSTRAINTS ON ACTIONS

We report the accuracies of our semantic parsers to show the difference among $\boldsymbol{\Psi}$ (Table 5). The functions in $\boldsymbol{\Psi}$ are listed in decreasing order of the number of applied constraints: $\Psi^{\text{HYBR}}$, $\Psi^{\text{TYPE}}$, $\Psi^{\text{TYPE}-}$ and $\Psi^{\text{NONE}}$. In the same order, the size of the action set $\Psi(r(\boldsymbol{a}^{\star}))$ for a function $\Psi \in \boldsymbol{\Psi}$ increases (Eq. 4). Therefore, a function $\Psi \in \boldsymbol{\Psi}$ with more constraints results in smaller search space, which is the set of all complete action sequences. Since the constraints reject actions that leads to an incorrect logical form, a search algorithm benefits from the small search space.

In particular, candidate expressions, which $\Psi^{\text{HYBR}}$ uses, made the biggest contribution to the improvement in accuracy. Candidate expressions are effective when a knowledge component is differently represented in an utterance $x$ since neural networks cannot correctly remember all KB components. For example, in Figure 1, the relation `"country of citizenship"`, which is a KB component, is represented as *"a citizen of"* in the utterance $x$, where the candidate expressions of the node class `query-rel-qualifier` can guide a semantic parser to generate the relation.

Although the effect of union types and other types was small, the type constraints consistently increased accuracies. Union types distinguish among actions that generate `(nlt *)` nodes, so the union types are useful for node classes (e.g., `constant-number`) that do not have candidate expressions due to their unlimited number of possible instances. Other types, which construct compositional structures or atomic units, are effective when the number of training examples is small. The type constraints would be useful for weakly-supervised learning where a semantic parser searches for an action sequence $\boldsymbol{a}$ whose denotation $[\![l(r(\boldsymbol{a}))]\!]_k$ equals the gold denotation $y$ during training (Liang et al., 2011; Krishnamurthy et al., 2017; Dasigi et al., 2019). Applying our grammar to weakly-supervised semantic parsing would be interesting future work (Appendix C).

### 7.2 EFFECT OF CANDIDATE EXPRESSIONS FOR EACH NODE CLASS

We report decreases in accuracies when candidate expressions for specific node classes were not used (Table 6). The candidate expressions for all the node classes contributed to accuracies unless the number of training examples was large. In particular, the node class `keyword-entity`, which has the most candidate expressions (Table 3), contributed the most to accuracies. Other node classes, such as `keyword-concept`, `keyword-relation` and `keyword-attribute-string` have many fewer candidate expressions than `keyword-entity`, but they also contributed to accuracies. The contributions to accuracies would increase when the KB becomes larger and the node classes have more candidate expressions.

Table 6: Reduced accuracies on $D^{\text{VAL}}$ when candidate expressions for specific node classes were not used.

| Unused class name | Number of training examples (percentage) | | | | | | |
|---|---|---|---|---|---|---|---|
| | 94 (0.1 %) | 283 (0.3 %) | 944 (1 %) | 2.83k (3 %) | 9.44k (10 %) | 28.3k (30 %) | 94.4k (100 %) |
| keyword-concept | 0.67 | 1.53 | 1.02 | 0.90 | 0.36 | 0.15 | 0.06 |
| keyword-entity | 0.71 | 1.29 | 1.26 | 1.24 | 1.16 | 0.95 | 0.65 |
| keyword-relation | 0.75 | 0.66 | 0.45 | 0.15 | 0.14 | 0.07 | 0.03 |
| keyword-attribute-string | 0.90 | 0.86 | 0.62 | 0.57 | 0.31 | 0.15 | 0.13 |
| keyword-attribute-number | 0.82 | 0.94 | 0.21 | 0.19 | 0.08 | 0.01 | 0.01 |
| keyword-attribute-time | 0.31 | 0.08 | 0.03 | 0.03 | 0.02 | 0.02 | 0.00 |
| keyword-qualifier-string | 0.03 | 0.03 | 0.03 | 0.02 | 0.03 | 0.02 | 0.01 |
| keyword-qualifier-number | 0.03 | 0.03 | 0.00 | 0.01 | 0.01 | 0.01 | 0.00 |
| keyword-qualifier-time | 0.01 | 0.07 | $-0.01$ | 0.01 | 0.00 | 0.00 | 0.00 |
| constant-unit | 0.07 | 0.33 | 0.03 | 0.02 | 0.03 | 0.01 | 0.00 |
| All | 4.01 | 5.41 | 3.52 | 3.09 | 2.09 | 1.37 | 0.88 |

## 8 RELATED WORK

We follow Yin & Neubig (2017); Krishnamurthy et al. (2017) whose grammars define actions as production rules that build well-typed formal representations such as logical forms or abstract syntax trees. The grammars can be designed for complex syntax, and be applied for various logical form languages (Yin & Neubig, 2018; Yin et al., 2018; Guo et al., 2019; Dasigi et al., 2019; Wang et al., 2020; Gupta et al., 2021; Cao et al., 2021b; Chen et al., 2021b). We further enhance the grammars with sub-type inference, union types (Section 3) and candidate expressions (Section 4) for flexible designs and to reduce search space.

There have been constrained decoding methods for semantic parsers based on seq2seq PLMs or LLMs. Scholak et al. (2021) use an incremental parser to filter hypotheses by examining the top-$k$ tokens with the highest prediction probabilities for each hypothesis. In contrast, our method instantly retrieves all valid actions $\Psi^{\text{HYBR}}(r(\boldsymbol{a}_{1:t-1}))$ for each time step $t$ during parsing. Shu et al. (2022) use constraints to decode operators into valid positions in logical forms, and to decode two categories of predicates by using two respective trie data structures. Their constrained decoding method can be formulated in our grammar framework with (1) three types to distinguish between operators, two categories of predicates, and (2) candidate expressions for the two categories. Therefore, our grammar framework is a generalization of their constrained decoding method. Wu et al. (2021); Shin et al. (2021); Poesia et al. (2022); Wang et al. (2023) developed grammar-based decoding methods for PLMs or LLMs with a few training examples. We briefly describes the idea to apply our grammar to the constrained decoding method by Shin et al. (2021) in Appendix D.

Bottom-up parsing has also been applied to neural semantic parsers, and it uses constrained decoding. Rubin & Berant (2021) define production rules for relational algebra (Codd, 1970), and they apply the production rules to compose logical forms in a bottom-up manner. Liang et al. (2017; 2018); Yin et al. (2020); Gu & Su (2022); Gu et al. (2023) execute sub-logical forms during inference, then use the constraints on the execution results. This approach can filter out meaningless logical forms that are valid with respect to types. However, executing sub-logical forms during inference requires non-trivial computational cost.

## 9 CONCLUSION

We present a grammar augmented with candidate expressions for semantic parsing on a large KB with a seq2seq PLM. Our grammar has a scalable and efficient design that incorporates both various types and many candidate expressions for a large KB. The grammar guides a semantic parser to construct compositional structures by using types, and to generate KB components by using candidate expressions. We experiment on the KQAPRO benchmark, and our semantic parsers achieved higher accuracies than previous work. In particular, semantic parsing with candidate expressions established state-of-the-art performance on KQAPRO.

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

# APPENDIX

We describe our grammar in more detail in Appendix A, the optimization algorithms that enhance the speed of constrained decoding in Appendix B, the application of our grammar to weakly-supervised learning in Appendix C, the combination of our grammar with Earley's algorithm (Earley, 1970) in Appendix D, and qualitative analysis in Appendix E.

## A   GRAMMAR DETAILS

### A.1   TYPES

The node classes in our grammar have return types and parameter types as properties, and the types have sub-type relations (Section 3) (Figure 3).

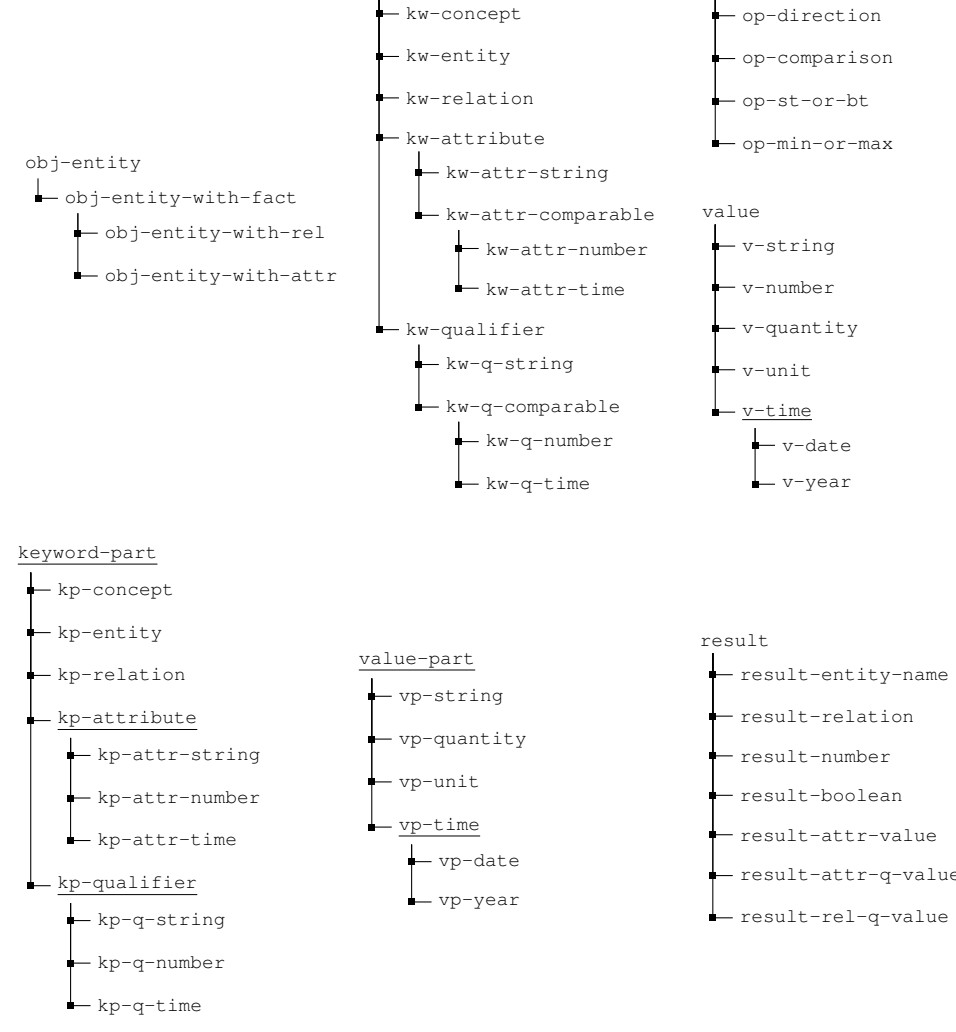

Figure 3: Type hierarchies. The types that are underlined are used only to group their sub-types without being used as return types or parameter types of node classes.

Table 7: Example of converting an intermediate representation to the corresponding logical form. For each row, a sub-intermediate representation that is written in bold with a specific color is converted to a sub-logical that is underlined with the color in the next row. Since the node class `find` has the same expression as its intermediate representation and as its logical form, the step $5 - 6$ makes no difference.

| Step | Intermediate representation → Logical form |
|---|---|
| 0 | `(program (query-rel-qualifier `**`(keyword-relation (nlt ·country) (nlt ·of)`**
        **`(nlt ·citizenship) reduce)`**
    **`(keyword-qualifier-time (nlt ·start) (nlt ·time) reduce)`**
    `(find `**`(keyword-entity (nlt ·Cary) (nlt ·Grant) reduce))`**
    `(find `**`(keyword-entity (nlt ·United) (nlt ·States)`**
        **`(nlt ·of) (nlt ·America) reduce))`** |
| $1 - 4$ | `(program (query-rel-qualifier ``"country of citizenship"`` ``"start time"`
    `(find ``"Cary Grant"``) (find ``"United States of America"``)))` |
| $5 - 6$ | `(program `**`(query-rel-qualifier "country of citizenship" "start time"`**
    **`(find "Cary Grant") (find "United States of America"))`**`)` |
| $7 - 8$ | **`(program (query-rel-qualifier (find "Cary Grant") (find "United States of America")`**
    **`"country of citizenship" "start time" ))`** |
| 9 | `(query-rel-qualifier (find "Cary Grant") (find "United States of America")`
    `"country of citizenship" "start time" )` |

## A.2 LOGICAL FORM TEMPLATES

An intermediate representation can be converted to a logical form by using logical form templates (Tables 1, 10 and 11). The conversion process is to apply the logical form template of each node to the intermediate representation of the node in a bottom-up manner (Table 7). The logical form template can be designed for various formats, such as S-expressions Krishnamurthy et al. (2017) or abstract syntax trees Yin & Neubig (2017). In our application to KQAPRO, a logical form is an S-expression for KBQA.

In the implementation of our grammar, (1) all actions have logical form templates that are closely related to Python code, and (2) many actions have logical form templates for simplified expressions. We use two keywords, `default` and `visual`, to separate the two groups of logical form templates (Figure 4). The `default` templates make logical forms that can be converted to the Python code for KBQA by using a transpiler (Odendahl, 2019). The Python code from a logical form is a function that takes a KB and return a denotation, where the function is a `lambda` expression and the KB is the argument `context`. In contrast, the `visual` templates make concise logical forms, which are used throughout this paper.

Since a logical form is constructed from templates rather than directly from actions, the order of the parameter types of an action is customizable. We customize the order of the parameter types of an action to put off constructing a sub-intermediate representation that requires relatively many actions. For example, some actions, such as `filter-number`, have `obj-entity` as a parameter type, and an intermediate representation for `obj-entity` has relatively many nodes; therefore, the actions put `obj-entity` as the last parameter type. The customized order of parameter types reduces the average difference in time steps between the actions that have parent-child relations; we assume that a seq2seq model benefits from the reduced distances between the actions, which are located in sequences.

## A.3 NODE CLASSES

Our grammar define node classes for the actions in $\mathcal{A}^{\text{COM}}$ and $\mathcal{A}^{\text{NLT}}$. The node classes for the actions in $\mathcal{A}^{\text{COM}}$ are manually specified (Tables 10 and 11). The node classes for the actions in $\mathcal{A}^{\text{NLT}}$ are converted from the natural language tokens of a seq2seq PLM by using a few rules.

The key differences between `(nlt *)` node classes are return types. To assign a proper return type to the `(nlt *)` node class from its natural language token, we use a few rules. First, the type of an `(nlt *)` node class is a union type that include the following types by default: `kp-concept`, `kp-entity`, `kp-relation`, `kp-attr-string`, `kp-attr-number`, `kp-attr-time`,

```
(define-action
  :name 'program
  :act-type 'result
  :param-types '(result)
  :expr-dict (mapkv :default $(lambda (context)
                                (postprocess-denotation {0}))
                    :visual "{0}")
  :starting True)

(define-action
  :name 'query-rel-qualifier
  :act-type 'result-rel-q-value
  :param-types '(kw-relation kw-qualifier obj-entity obj-entity)
  :expr-dict (mapkv :default $(context.QueryRelationQualifier {2} {3} {0} {1})
                    :visual $(query-rel-qualifier {2} {3} {0} {1})))

(define-action
  :name 'keyword-relation
  :act-type 'kw-relation
  :param-types '(kp-relation &rest kp-relation)
  :arg-candidate (retrieve '(candidate kw-relation))
  :expr-dict (mapkv :default (retrieve '(function concat-parts))))

(define-meta-action
  :meta-name 'nl-token
  :meta-params '(token)
  :name-fn (retrieve '(name nl-token))
  :expr-dict-fn (lambda (token)
                  (mapkv :default token))
  :param-types '())
```

Figure 4: Part of code for our grammar. In the code, an action for a node class is defined by `define-action`. There is discrepancies in terminology between the code and our grammar; `act-type` means a return type, `param-types` means parameter types, `expr-dict` means logical form templates, and `arg-candidate` means a function that uses candidate expressions for the node class. In addition, the code defines the meta-action for the meta-node class `nl-token`, which corresponds to the `nlt` symbol in `(nlt *)` node classes. The meta-node class and the tokens from a seq2seq PLM create all actions for `(nlt *)` node classes.

`kp-q-string`, `kp-q-number`, `kp-q-time`, `vp-string` and `vp-unit`. Second, the return type of the node class additionally include `vp-quantity`, `vp-date` or `vp-year` if the natural language token of the node class is a number or a special character for the types. For example, the return type of `(nlt ·7)` includes the three additional types, and the return type of `(nlt ·.)` includes `vp-quantity` since the period character (.) is used for rational numbers (e.g., 3.14).

**Algorithm 1** Method to compute a set that consists of either all valid actions or all invalid actions. The input values are an incomplete action sequence $\boldsymbol{a}^\star$ and a cache memory $M$. The return value is either $\langle \Psi^{\text{HYBR}}(r(\boldsymbol{a}^\star)), \text{True} \rangle$ or $\langle \mathcal{A} - \Psi^{\text{HYBR}}(r(\boldsymbol{a}^\star)), \text{False} \rangle$

---

**procedure** COMPUTEACTIONSVALIDNESS($\boldsymbol{a}^\star, M$)
    **if** HASCANDEXPR($\rho(\nu(r(\boldsymbol{a}^\star)))$) **then**
        **return** $\langle \Psi^{\text{CAND}}(r(\boldsymbol{a}^\star)), \text{True} \rangle$               $\triangleright$ Assume $|\Psi^{\text{CAND}}(r(\boldsymbol{a}^\star))| < \frac{1}{2}|\mathcal{A}|$
    **else**
        **if** TYPE($\nu(r(\boldsymbol{a}^\star))$) $\notin$ KEYS($M$) **then**
            **if** $|\Psi^{\text{TYPE}}(r(\boldsymbol{a}^\star))| < \frac{1}{2}|\mathcal{A}|$ **then**
                $M[\text{TYPE}(\nu(r(\boldsymbol{a}^\star)))] \leftarrow \langle \Psi^{\text{TYPE}}(r(\boldsymbol{a}^\star)), \text{True} \rangle$
            **else**
                $M[\text{TYPE}(\nu(r(\boldsymbol{a}^\star)))] \leftarrow \langle \mathcal{A} - \Psi^{\text{TYPE}}(r(\boldsymbol{a}^\star)), \text{False} \rangle$
            **end if**
        **end if**
        **return** $M[\text{TYPE}(\nu(r(\boldsymbol{a}^\star)))]$
    **end if**
**end procedure**

---

**Algorithm 2** Method to compute a scoring mask for valid actions. The input values are a batch $B$ of incomplete action sequences and a cache memory $M$. The return value is a scoring mask $S$.

---

**procedure** COMPUTESCORINGMASK($B, M$)
    $S \leftarrow$ a tensor with the size of $|B| \times |\mathcal{A}|$
    **for** $i \leftarrow 1$ to $|B|$ **do**
        **for** $j \leftarrow 1$ to $|\mathcal{A}|$ **do**
            $S_{i,j} \leftarrow -\infty$                          $\triangleright$ By GPUs
        **end for**
    **end for**
    **for** $k \leftarrow 1$ to $|B|$ **do**
        $\boldsymbol{a}^\star \leftarrow B_k$
        $\langle C, v \rangle \leftarrow$ COMPUTEACTIONSVALIDNESS($\boldsymbol{a}^\star, M$)          $\triangleright$ Mostly, $|C| < |\mathcal{A} - C|$
        **if** $v$ **then**
            **for** $a \in C$ **do**
                $S_{k,\text{ID}(a)} \leftarrow 0$                   $\triangleright$ By CPUs
            **end for**
        **else**
            **for** $i \leftarrow 1$ to $|\mathcal{A}|$ **do**
                $S_{k,i} \leftarrow 0$                       $\triangleright$ By GPUs
            **end for**
            **for** $a \in C$ **do**
                $S_{k,\text{ID}(a)} \leftarrow -\infty$             $\triangleright$ By CPUs
            **end for**
        **end if**
    **end for**
    **return** $S$
**end procedure**

---

## B    DECODING SPEED OPTIMIZATION

For $\Psi \in \{\Psi^{\text{HYBR}}, \Psi^{\text{TYPE}}, \Psi^{\text{TYPE}-}\}$, we observed that $|\Psi(r(\boldsymbol{a}^\star))|$ is usually very small or large. For example, when the leftmost non-terminal $\nu(r(\boldsymbol{a}^\star))$ has the type `op-direction`, $\Psi(r(\boldsymbol{a}^\star))$ includes actions that produce `'forward` and `'backward`, then $|\Psi(r(\boldsymbol{a}^\star))| = 2$. For another example, when the leftmost non-terminal $\nu(r(\boldsymbol{a}^\star))$ has the type `vp-string`, $\Psi(r(\boldsymbol{a}^\star)) = \mathcal{A}^{\text{NLT}}$, then $|\Psi(r(\boldsymbol{a}^\star))| = 50,260$.

We devise algorithms that enhance the speed of constrained decoding by patterns of the size $|\Psi(r(\boldsymbol{a}^\star))|$ (Algorithms 1 and 2). Algorithm 1 retrieves either (1) a set of valid actions from trie

Table 8: Average time to decode an output sequence by greedy search. When optimization is disabled, Algorithm 1 doesn't consider the size $|\Psi^{\text{TYPE}}(r(\boldsymbol{a}^\star))|$, then the cache memory $M$ is only used to store $\langle\Psi^{\text{TYPE}}(r(\boldsymbol{a}^\star)), \text{True}\rangle$, and the return value is always $\langle\Psi^{\text{HYBR}}(r(\boldsymbol{a}^\star)), \text{True}\rangle$.

| Model | Optimization | Time (ms) | |
|---|---|---|---|
| | | Batch size = 1 | Batch size = 64 |
| BART KoPL (Cao et al., 2022) | N/A | 117.8 | 4.5 |
| Ours with $\Psi^{\text{NONE}}$ | N/A | 101.7 | 3.8 |
| Ours with $\Psi^{\text{TYPE}-}$ | ✓ | 110.3 | 10.0 |
| Ours with $\Psi^{\text{TYPE}}$ | ✓ | 110.0 | 10.0 |
| Ours with $\Psi^{\text{HYBR}}$ | ✓ | 110.7 | 10.2 |
| Ours with $\Psi^{\text{TYPE}-}$ | ✗ | 140.9 | 39.3 |
| Ours with $\Psi^{\text{TYPE}}$ | ✗ | 138.9 | 37.5 |
| Ours with $\Psi^{\text{HYBR}}$ | ✗ | 114.9 | 14.7 |

data structures for candidate expressions, or (2) a set of valid actions or a set of invalid actions from a cache memory for types. Algorithm 2 takes the retrieved set of actions as an input, then computes a scoring mask, where the usage of CPUs are minimized. The algorithms generalize the `PrefixConstrainedLogitsProcessor` method (Cao et al., 2021a), which is implemented in the transformers library (Wolf et al., 2020).

We also measured the average decoding time with different settings (Table 8). The result shows that the optimization algorithms achieved meaningful decrease in decoding time. In particular, the effect of the algorithms is noticeable when the batch size is large. This means that our algorithms greatly reduce the overhead occurred from our constraints, then search algorithms such as beam search, which perform concurrent operations, benefit from the reduction in the overhead. In addition, $\Psi^{\text{TYPE}-}$ and $\Psi^{\text{TYPE}}$ greatly benefit from the algorithms, since $\Psi^{\text{TYPE}-}(r(\boldsymbol{a}^\star))$ and $\Psi^{\text{TYPE}}(r(\boldsymbol{a}^\star))$ have many actions in $\mathcal{A}^{\text{NLT}}$ as valid actions due to the absence of candidate expressions.

We note that the decoding time is proportional to the length of an output sequence. Since, ours and BART KoPL respectively have 28.8 and 35.1 as the average lengths of output sequences, ours with $\Psi^{\text{NONE}}$ is slightly faster than BART KoPL, although both don't use any constraint. BART KoPL tokenizes symbols, such as function names, then its output sequences include slight more tokens. If sub-type inference (Section 3) is not applied to ours, we should additionally introduce actions that convert a non-terminal with a super type to a non-terminal with sub-type; an example action is `<result>` → `<result-rel-q-value>`. The average lengths of output sequences of ours without sub-type inference is 32.8, then its decoding slightly gets slower.

## C APPLICATION TO WEAKLY-SUPERVISED LEARNING

Semantic parsers can learn to predict logical forms from weak supervision of gold denotations, which correspond to gold answers in the task of question answering. The learning process repeats two steps: (1) finding consistent logical forms, which produce gold denotations, by searching with the current parameters of a semantic parser, then (2) optimizing the parameters by learning from the consistent logical forms (Liang et al., 2011; Berant et al., 2013; Krishnamurthy et al., 2017; Guu et al., 2017; Liang et al., 2017; Goldman et al., 2018; Dasigi et al., 2019; Liang et al., 2018). However, weakly-supervised semantic parsing is less addressed for seq2seq PLMs, and existing work (Wolfson et al., 2022) also does not use a semantic parser for search during training.

Our grammar can guide a weakly-supervised semantic parser based on seq2seq PLMs to find a consistent intermediate representation $r(\boldsymbol{a})$, whose denotation $[\![l(r(\boldsymbol{a}))]\!]_k$ is identical with a gold denotation $y$, when given an utterance $x$. Once consistent intermediate representations are found, our semantic parser can learn from the action sequences of the intermediate representations by maximizing marginal log likelihood (Krishnamurthy et al., 2017; Dasigi et al., 2019) or expected reward (Liang et al., 2017; 2018). These search and optimization steps are repeated, then the denotation accuracy of the semantic parser gradually increases.

Table 9: Oracle denotation accuracies on $D^{\text{VAL}}$. For all the accuracies, the same model, which is trained from 0.1% (98 examples) of $D^{\text{TRAIN}}$, is evaluated with different functions in $\boldsymbol{\Psi}$ and with different beam sizes. In the experiment, beam search with beam size $K$ finds $K$ intermediate representations with the approximately highest likelihoods. When the bean size is 1, both oracle denotation accuracy and denotation accuracy are same.

| $\Psi \in \boldsymbol{\Psi}$ | Constraints | | | Beam size | | | | |
|---|---|---|---|---|---|---|---|---|
| | Cand. expr. | Union types | Types | 1 | 4 | 8 | 12 | 16 |
| $\Psi^{\text{HYBR}}$ | ✓ | ✓ | ✓ | 35.36 | 49.72 | 54.09 | 56.49 | 58.01 |
| $\Psi^{\text{TYPE}}$ | ✗ | ✓ | ✓ | 31.36 | 42.41 | 46.28 | 48.05 | 49.28 |
| $\Psi^{\text{TYPE}-}$ | ✗ | ✗ | ✓ | 31.29 | 42.32 | 46.22 | 48.05 | 49.23 |
| $\Psi^{\text{NONE}}$ | ✗ | ✗ | ✗ | 28.44 | 37.34 | 40.84 | 42.65 | 44.00 |

To verify the effectiveness of our grammar in weakly-supervised semantic parsing, we conducted a preliminary experiment where an insufficiently trained semantic parser performs beam search to find consistent intermediate representations under the guidance of a function $\Psi \in \boldsymbol{\Psi}$ (Table 9). In the experiment, we measured oracle denotation accuracy, which is the fraction of examples where a search algorithm, which uses $p_\theta(\boldsymbol{a} \mid x)$, finds at least one consistent intermediate representation. The experimental result shows that constraints by types and especially by candidate expressions increase oracle denotation accuracies.

## D    COMBINATION WITH EARLEY'S ALGORITHM

Shin et al. (2021) combine Synchronous CFGs (SCFGs) with Earley's Algorithm (Earley, 1970) where PLMs or LLMs generate canonical utterances but SCFGs also parse logical forms. This enable PLMs or LLMs to generate representations in a formal language and synchronously track their counterparts in another formal language. If the decoding method internally tracks our intermediate representations, the constraints by candidate expressions can be applied.

## E    QUALITATIVE ANALYSIS

We compare the results of semantic parsing with $\Psi^{\text{HYBR}}$ and with $\Psi^{\text{TYPE}}$ (Tables 12 to 14). For an utterance $x$, an intermediate representation $r(\boldsymbol{a})$ and a logical form $l(r(\boldsymbol{a}))$, we highlight the parts that correspond to a candidate expression. The highlighted part in $x$ has a different representation from that of the corresponding candidate expression. With the guidance of $\Psi^{\text{HYBR}}$, the highlighted parts in $r(\boldsymbol{a})$ and $l(r(\boldsymbol{a}))$ are valid KB components. In contrast, with the guidance of $\Psi^{\text{TYPE}}$, the highlighted parts in $r(\boldsymbol{a})$ and $l(r(\boldsymbol{a}))$ are invalid, since a semantic parser copies the parts from $x$ just as it is.

Table 10: First part of node classes for compositional structures and atomic units.

| Class name | Return type | Parameter types | Logical form template |
|---|---|---|---|
| program | result | [result] | @0 |
| all-entities | obj-entity | N/A | all-entities |
| find | obj-entity | [kw-entity] | (find @0) |
| filter-concept | obj-entity | [kw-concept obj-entity] | (filter-concept @1 @0) |
| filter-str | obj-entity-with-attr | [kw-attr-string v-string obj-entity] | (filter-str @2 @0 @1) |
| filter-number | obj-entity-with-attr | [kw-attr-number v-number op-comparison obj-entity] | (filter-number @3 @0 @1 @2) |
| filter-year | obj-entity-with-attr | [kw-attr-year v-year op-comparison obj-entity] | (filter-year @3 @0 @1 @2) |
| filter-date | obj-entity-with-attr | [kw-attr-date v-date op-comparison obj-entity] | (filter-date @3 @0 @1 @2) |
| relate | obj-entity-with-rel | [kw-relation op-direction obj-entity] | (relate @2 @0 @1) |
| op-eq | op-comparison | N/A | = |
| op-ne | op-comparison | N/A | != |
| op-lt | op-comparison | N/A | < |
| op-gt | op-comparison | N/A | > |
| direction-forward | op-direction | N/A | 'forward |
| direction-backward | op-direction | N/A | 'backward |
| q-filter-str | obj-entity-with-fact | [kw-q-string v-string obj-entity-with-fact] | (q-filter-str @2 @0 @1) |
| q-filter-number | obj-entity-with-fact | [kw-q-number v-number op-comparison obj-entity-with-fact] | (q-filter-number @3 @0 @1 @2) |
| q-filter-year | obj-entity-with-fact | [kw-q-year v-year op-comparison obj-entity-with-fact] | (q-filter-year @3 @0 @1 @2) |
| q-filter-date | obj-entity-with-fact | [kw-q-date v-date op-comparison obj-entity-with-fact] | (q-filter-date @3 @0 @1 @2) |
| intersect | obj-entity | [obj-entity obj-entity] | (intersect @0 @1) |
| union | obj-entity | [obj-entity obj-entity] | (union @0 @1) |
| count | result-number | [obj-entity] | (count @0) |
| select-between | result-entity-name | [kw-attr-comparable op-st-or-bt obj-entity obj-entity] | (select-between @2 @3 @0 @1) |
| select-among | result-entity-name | [kw-attr-comparable op-min-or-max obj-entity | (select-among @2 @0 @1) |
| op-st | op-st-or-bt | N/A | 'less |
| op-bt | op-st-or-bt | N/A | 'greater |
| op-min | op-min-or-max | N/A | 'min |
| op-max | op-min-or-max | N/A | 'max |

Table 11: Second part of node classes for compositional structures and atomic units.

| Class name | Return type | Parameter types | Logical form template |
|---|---|---|---|
| query-name | result-entity-name | [obj-entity] | (query-name @0) |
| query-attr | result-attr-value | [kw-attribute obj-entity] | (query-attr @0 @1) |
| query-attr-under-cond | result-attr-value | [kw-attribute kw-qualifier value obj-entity] | (query-attr-under-cond @3 @0 @1 @2) |
| query-relation | result-relation | [obj-entity obj-entity] | (query-relation @0 @1) |
| query-attr-qualifier | result-attr-q-value | [kw-attribute value kw-qualifier obj-entity ] | (query-attr-qualifier @3 @0 @1 @2) |
| query-rel-qualifier | result-rel-q-value | [kw-relation kw-qualifier obj-entity obj-entity] | (query-rel-qualifier @2 @3 @0 @1) |
| verify-str | result-boolean | [v-string result-attr-value] | (verify-str @1 @0) |
| verify-number | result-boolean | [v-number op-comparison result-attr-value] | (verify-number @2 @0 @1) |
| verify-year | result-boolean | [v-year op-comparison result-attr-value] | (verify-year @2 @0 @1) |
| verify-date | result-boolean | [v-date op-comparison result-attr-value] | (verify-date @2 @0 @1) |
| keyword-concept | kw-concept | [kp-concept &rest kp-concept] | #(concat @*) |
| keyword-entity | kw-entity | [kp-entity &rest kp-entity] | #(concat @*) |
| keyword-relation | kw-relation | [kp-relation &rest kp-relation] | #(concat @*) |
| keyword-attribute-string | kw-attr-string | [kp-attr-string &rest kp-attr-string] | #(concat @*) |
| keyword-attribute-number | kw-attr-number | [kp-attr-number &rest kp-attr-number] | #(concat @*) |
| keyword-attribute-time | kw-attr-time | [kp-attr-time &rest kp-attr-time] | #(concat @*) |
| keyword-qualifier-string | kw-q-string | [kp-q-string &rest kp-q-string] | #(concat @*) |
| keyword-qualifier-number | kw-q-number | [kp-q-number &rest kp-q-number] | #(concat @*) |
| keyword-qualifier-time | kw-q-time | [kp-q-time &rest kp-q-time] | #(concat @*) |
| constant-string | v-string | [vp-string &rest vp-string] | #(concat @*) |
| constant-year | v-year | [vp-year &rest vp-year] | #(concat @*) |
| constant-date | v-date | [vp-date &rest vp-date] | #(concat @*) |
| constant-number | v-number | [v-quantity v-unit] | #(concat-quantity-unit @*) |
| constant-quantity | v-quantity | [vp-quantity &rest vp-quantity] | #(raw-concat @*) |
| constant-unit | v-unit | [&rest vp-unit] | #(raw-concat @*) |

Table 12: Example of semantic parsing when $\Psi^{\text{CAND}}$ is effective for `keyword-concept`. The correct candidate expression is `game show` rather than `game`.

| | |
|---:|:---|
| $x:$ | *What game has more than 630 episodes and originally aired on NBC?* |
| $y:$ | The Price Is Right |
| $\Psi:$ | $\Psi^{\text{HYBR}}$ |
| $r(\boldsymbol{a}):$ | `(program (query-name (intersect (filter-concept (keyword-concept (nlt ·game) (nlt ·show) reduce) (filter-number (keyword-attribute-number (nlt ·number) (nlt ·of) (nlt ·episodes) reduce) (constant-number (constant-quantity (nlt ·630) reduce) (constant-unit reduce)) op-gt all-entities)) (filter-concept (keyword-concept (nlt ·game) (nlt ·show) reduce) (relate (keyword-relation (nlt ·original) (nlt ·network) reduce) direction-backward (find (keyword-entity (nlt ·NBC) reduce)))))))` |
| $l(r(\boldsymbol{a})):$ | `(query-name (intersect (filter-concept (filter-number all-entities "number of episodes" "630" >) "game show") (filter-concept (relate (find "NBC") "original network" 'backward) "game show")))` |
| $[\![l(r(\boldsymbol{a}))]\!]_k:$ | The Price Is Right |
| $\Psi:$ | $\Psi^{\text{TYPE}}$ |
| $r(\boldsymbol{a}):$ | `(program (query-name (intersect (filter-concept (keyword-concept (nlt ·game) reduce) (filter-number (keyword-attribute-number (nlt ·number) (nlt ·of) (nlt ·episodes) reduce) (constant-number (constant-quantity (nlt ·630) reduce) (constant-unit reduce)) op-gt all-entities)) (filter-concept (keyword-concept (nlt ·game) reduce) (relate (keyword-relation (nlt ·original) (nlt ·network) reduce) direction-backward (find (keyword-entity (nlt ·NBC) reduce)))))))` |
| $l(r(\boldsymbol{a})):$ | `(query-name (intersect (filter-concept (filter-number all-entities "number of episodes" "630" >) "game") (filter-concept (relate (find "NBC") "original network" 'backward) "game")))` |
| $[\![l(r(\boldsymbol{a}))]\!]_k:$ | N/A |

Table 13: Example of semantic parsing when $\Psi^{\text{CAND}}$ is effective for `keyword-entity`. The correct candidate expression is `Tilda Swinton` rather than `Tilde Swinton`.

| | |
|---:|:---|
| $x:$ | *What film has Tilde Swinton in the cast and was distributed by StudioCanal?* |
| $y:$ | Julia |
| $\Psi:$ | $\Psi^{\text{HYBR}}$ |
| $r(\boldsymbol{a}):$ | `(program (query-name (intersect (filter-concept (keyword-concept (nlt ·film) reduce) (relate (keyword-relation (nlt ·cast) (nlt ·member) reduce) direction-backward (find (keyword-entity (nlt ·T) (nlt ·ilda) (nlt ·Sw) (nlt ·inton) reduce)))) (filter-concept (keyword-concept (nlt ·film) reduce) (relate (keyword-relation (nlt ·distributor) reduce) direction-backward (find (keyword-entity (nlt ·Studio) (nlt ·Can) (nlt ·al) reduce)))))))` |
| $l(r(\boldsymbol{a})):$ | `(query-name (intersect (filter-concept (relate (find "Tilda Swinton") "cast member" 'backward) "film") (filter-concept (relate (find "StudioCanal") "distributor" 'backward) "film")))` |
| $[\![l(r(\boldsymbol{a}))]\!]_k:$ | Julia |
| $\Psi:$ | $\Psi^{\text{TYPE}}$ |
| $r(\boldsymbol{a}):$ | `(program (query-name (intersect (filter-concept (keyword-concept (nlt ·film) reduce) (relate (keyword-relation (nlt ·cast) (nlt ·member) reduce) direction-backward (find (keyword-entity (nlt ·T) (nlt ·ilde) (nlt ·Sw) (nlt ·inton) reduce)))) (filter-concept (keyword-concept (nlt ·film) reduce) (relate (keyword-relation (nlt ·distributor) reduce) direction-backward (find (keyword-entity (nlt ·Studio) (nlt ·Can) (nlt ·al) reduce)))))))` |
| $l(r(\boldsymbol{a})):$ | `(query-name (intersect (filter-concept (relate (find "Tilde Swinton") "cast member" 'backward) "film") (filter-concept (relate (find "StudioCanal") "distributor" 'backward) "film")))` |
| $[\![l(r(\boldsymbol{a}))]\!]_k:$ | N/A |

Table 14: Example of semantic parsing when $\Psi^{\text{CAND}}$ is effective for `keyword-relation`. The correct candidate expression is `ethnic group` rather than `ethnic community`.

| | |
|---|---|
| $x:$ | *How many historical countries use the Japanese yen as their currency or are an ethnic community of African Americans?* |
| $y:$ | 2 |
| $\Psi:$ | $\Psi^{\text{HYBR}}$ |
| $r(\boldsymbol{a}):$ | ```(program (count (union (filter-concept (keyword-concept (nlt ·historical) (nlt ·country) reduce) (relate (keyword-relation (nlt ·currency) reduce) direction-backward (find (keyword-entity (nlt ·Japanese) (nlt ·yen) reduce)))) (filter-concept (keyword-concept (nlt ·historical) (nlt ·country) reduce) (relate (keyword-relation (nlt ·ethnic) (nlt ·group) reduce) direction-backward (find (keyword-entity (nlt ·African) (nlt ·Americans) reduce)))))))``` |
| $l(r(\boldsymbol{a})):$ | ```(count (union (filter-concept (relate (find "Japanese yen") "currency" 'backward) "historical country") (filter-concept (relate (find "African Americans") "ethnic group" 'backward) "historical country")))``` |
| $[\![l(r(\boldsymbol{a}))]\!]_k:$ | 2 |
| $\Psi:$ | $\Psi^{\text{TYPE}}$ |
| $r(\boldsymbol{a}):$ | ```(program (count (union (filter-concept (keyword-concept (nlt ·historical) (nlt ·country) reduce) (relate (keyword-relation (nlt ·currency) reduce) direction-backward (find (keyword-entity (nlt ·Japanese) (nlt ·yen) reduce)))) (filter-concept (keyword-concept (nlt ·historical) (nlt ·country) reduce) (relate (keyword-relation (nlt ·ethnic) (nlt ·community) reduce) direction-backward (find (keyword-entity (nlt ·African) (nlt ·Americans) reduce)))))))``` |
| $l(r(\boldsymbol{a})):$ | ```(count (union (filter-concept (relate (find "Japanese yen") "currency" 'backward) "historical country") (filter-concept (relate (find "African Americans") "ethnic community" 'backward) "historical country")))``` |
| $[\![l(r(\boldsymbol{a}))]\!]_k:$ | 1 |

