# OpenReview forum: "Semantic Parsing with Candidate Expressions for Knowledge Base Question Answering"
_ICLR.cc/2024/Conference — Submitted to ICLR 2024_

### Official Review · Reviewer_iWQY · 2023-10-19

**Soundness:** 3 good
**Presentation:** 3 good
**Contribution:** 2 fair
**Rating:** 6
**Confidence:** 3

**Summary:**

The paper presents a constraint decoding method for semantic parsing. The paper leverages pretrained neural models and tried to narrow down decoding search space by defining IR, type, candidate expressions (whitelisting) to improve semantic parser performance. It goes into details telling the reason why constraint decoding is needed and how it's implemented. Experiment is conducted against KB, and results looks good. The authors put together a complex neural semantic parsing pipeline and proposed a solution for the problem of constrained decoding for non-terminal nodes in structured decoding. On `multi-hop` and `qualifier` test sets, there are decent improvements.

**Strengths:**

- The constraint decoding approach shows decent improvements in experiments.
- The authors designed an intermediate representation system based on s-expression which is more generic comparing with other constrained decoding method like Picard. The candidate expression is built as a trie tree structure. Comparing with previous work, this work expands the constraints to internal nodes (non-terminal nodes) which would be helpful to narrow down search space.
- The writing is clean and easy to follow

**Weaknesses:**

- This is a relatively incremental work. Constraint decoding is not a new idea which is actually widely used in semantic parsing field. The trie tree structure for candidate expression decoding is also not new (similar to GENRE). The novel part is to expand it to non-terminal node.
- In the era of LLMs, the improvement seems marginal comparing with using large LMs and with more training data.

**Questions:**

- Comparing with using production rules etc, have you tried unstructured decoding method? Maybe this should be a good baseline.
- Clearly with more data, the constrained decoding doesn't help much. Have you explored data augmentation?

---

> ### Author Response · Authors · 2023-11-14
> **Response to Reviewer iWQY**
>
> We sincerely thank you for spending your precious time to review our paper.
>
> **Response to Questions**
>
> We first reply to your questions.
>
> >Comparing with using production rules etc, have you tried unstructured decoding method? Maybe this should be a good baseline.
>
> BART KoPL [1] can be a good baseline that uses unstructured decoding.
> BART KoPL predicts logical forms written in KoPL, which is linearized in postfix representations.
> An example output sequence of BART KoPL is
> ```
> <s>FindAll <func> FilterStr <arg> official website <arg> http://www.thesiege.com/ <func> FilterConcept <arg> visual artwork <func> QueryAttrQualifier <arg> publication date <arg> 1999-01-21 <arg> place of publication</s>
> ```
> which are represented in our model as
> ```
> (query-attr-qualifier (filter-concept (filter-str all-entities "official website" "http://www.thesiege.com/")
>                                       "visual artwork")
>                       "publication date" "1999-01-21" "place of publication")
> ```
>
> Unlike ours, BART KoPL splits symbols (e.g. FindAll or FilterStr) into tokens (e.g. Find, All, Filter, Str), then the model has slightly longer output sequences.
>
> We also analyzed the speed of ours and BART KoPL in the reply to the reviewer pMus.
>
> >Clearly with more data, the constrained decoding doesn't help much.
>
> We think that constraints on the syntax don't much enhance performance with a large training set,
> since the limited number of syntactic patterns are repeated in the training set.
> As your comment, experiment results also show that candidate expressions are less effective with a large training set.
> However, we still think that candidate expressions are useful, especially when inferring KB components (e.g. entities or relations) that are unseen during training.
> Similarly with GENRE [4], candidate expressions would guide semantic parsers when a KB is updated after training (e.g. new entities are added).
>
> >Have you explored data augmentation?
>
> KQAPRO [1] was semi-automatically constructed in two steps [9]; (1) automatically generating pairs of canonical utterances and logical forms by using SCFG, and (2) paraphrasing the canonical utterances by crowdsourcing.
> In addition, zero-shot semantic parsing [10] showed decent performance without manually annotated training and validation sets.
> Another interesting work generates syntactic pairs of canonical utterances and logical forms during weakly-supervised learning [11].
>
> The core of data synthesis method is using SCFGs, which consist of pairs of production rules that generate canonical utterances and logical forms.
> If we additionally add templates for canonical utterances to all node classes, our grammar can be considered as a SCFG, where a complete intermediate representation can generate a pairs of a canonical utterance and a logical form from two kinds of templates.
>
> We think that grammar-based data synthesis is a good research area for semantic parsing.
>
> **Additional Explanation**
>
> Please let us explain more about our paper with respect to your comments.
>
> >In the era of LLMs, the improvement seems marginal comparing with using large LMs and with more training data.
>
> As your comment, LLMs can achieve high performance with many training examples.
> In addition, LLMs such as GPT3 enables in-context learning, where our method cannot be directly applied.
> However, we've recently found interesting semantic parsers that generate tokens (not production rules) but contrain tokens with respect to SCFGs [2,3].
> Following [3], our method can be combined with Earley’s algorithm [12], where our intermediate representations can replace dotted production rules.
> In our revision, we'll describe [2,3] and the idea of applying our method to [2,3].
>
> **TODO**
>
> We're going to update our paper during the discussion period.
>
> - Constrained decoding methods [2,3] for pretrained seq2seq will be added to "Related work".
>   - We will also briefly describe how our method can be combined with [2,3].
>
> **Gratitude**
>
> We again greatly thank you for your reviews and feedback.
> If you have any more comments or questions, please feel free.
>
> **Reference**
>
> [1] KQA Pro: A Dataset with Explicit Compositional Programs for Complex Question Answering over Knowledge Base
> [2] From Paraphrasing to Semantic Parsing: Unsupervised Semantic Parsing via Synchronous Semantic Decoding
> [3] Constrained Language Models Yield Few-Shot Semantic Parsers
> [4] Autoregressive Entity Retrieval
> [9] Building a Semantic Parser Overnight
> [10] On The Ingredients of an Effective Zero-shot Semantic Parser
> [11] Weakly Supervised Semantic Parsing by Learning from Mistakes
> [12] An Efficient Context-Free Parsing Algorithm

---

### Official Review · Reviewer_4fJL · 2023-11-01

**Soundness:** 3 good
**Presentation:** 3 good
**Contribution:** 1 poor
**Rating:** 3
**Confidence:** 4

**Summary:**

This paper proposes an approach to semantic parsing over knowledge base using candidate expressions, which improves the accuracy of semantic parsers on knowledge bases. They evaluate their approach on KQAPro and show that it outperforms other state-of-the-art methods. The authors also conduct an ablation study to analyze the contribution of candidate expressions to the performance of the semantic parser. Overall, the paper's contributions include a new approach to semantic parsing using candidate expressions, and an evaluation of the effectiveness of candidate expressions in improving the accuracy of semantic parsers.

**Strengths:**

- This paper provides a clear and detailed explanation of their proposed approach, including the grammar and inference algorithm used to generate candidate expressions.
- This paper conducts an ablation study to analyze the contribution of candidate expressions to the performance of the semantic parser. This analysis provides insights into the effectiveness of candidate expressions and how they can be used to improve the accuracy of semantic parsers.
- The paper is of high quality, with clear and well-organized writing and thorough experimental evaluation.

**Weaknesses:**

The major weakness of this work is its incremental contribution over previous grammar-based methods like [1] and [2]. In particular, [2] uses similar grammars for knowledge base question answering. It is unclear what new techniques or insights this work adds beyond existing grammar-based approaches for this task. To strengthen the paper, the authors could focus more on novel grammar designs or representations that improve performance.

Additionally, the experimental validation is limited to a single dataset (KQAPro). Testing the approach on an additional challenging dataset like GrailQA [3] would better verify effectiveness and generalization. With only one dataset, it is hard to determine if the performance gains are dataset-specific or represent a robust advancement for grammar-based decoding. Expanding the experimental evaluation would make the results more convincing.

In summary, clearly situating the contributions relative to prior grammar-based work and testing across more datasets could help address concerns around novelty and experimental validation. Focusing on where this work provides specific technical innovations or new insights would strengthen the paper.

[1]. A Syntactic Neural Model for General-Purpose Code Generation
[2]. ReTraCk: A Flexible and Efficient Framework for Knowledge Base Question Answering
[3]. https://dki-lab.github.io/GrailQA/

**Questions:**

N/A

---

> ### Author Response · Authors · 2023-11-14
> **Response to Reviewer 4fJL (Part 1)**
>
> We sincerely thank you for spending your precious time to review our paper.
>
> **Additional Explanation**
>
> Please let us explain more about our paper with respect to your comments.
>
> >The major weakness of this work is its incremental contribution over previous grammar-based methods like [1] and [2]. In particular, [2] uses similar grammars for knowledge base question answering.
>
> Thank you for letting us know ReTraCk [13], which also uses grammar-based decoding [5,6,7] and which is also applied to KBQA.
> We checked ReTraCk and found that ReTraCk has the "Checker" module which uses constraints that are specialized for KBQA.
> We think ReTraCk has a good modular design and robust constraints.
> We'll cite ReTraCk in our revision.
>
> However, please let us explain the difference between ours and previous semantic parsers, including ReTraCk, which use grammar-based decoding.
> The purpose of our paper to apply constraints based on a grammar to a pretrained __seq2seq__ models, such as BART or T5, which have pretrained __decoders with tokenizers__.
> To exploit a pretrained seq2seq model, the output sequences should be composed of tokens that are also used during pretraining.
> In contrast, previous semantic parsers [7,8,13] (e.g. ReTraCk) that have constraints on KBs (or structured data) treat KB components (e.g. entities or relations) as atomic units, which are not further split into tokens.
> Therefore, the semantic parsers with the sophisticated constraints couldn't benefit from pretrained decoders, so the models chose custom LSTMs as their decoders.
>
> Meanwhile, semantic parsers based on pretrained seq2seq models have shown good performance [14,15,16], but they haven't used grammar-based decoding.
> Therefore, we applied grammar-based decoding [5,7] to a pretrained seq2seq model, and introduced candidate expressions which enable constraints on KBs for the pretrained seq2seq model.
> The current application of candidate expressions with trie data structures is similar to "Instance-level Checking" of ReTraCk, but our constraints are for pretrained seq2seq models.
> We also thinks that candidate expressions can be applied to other constraints, such as "Ontology-level Checking" of ReTraCk.
>
> In addition, we've recently found interesting semantic parsers that generate tokens (not production rules) but constrain tokens with respect to SCFGs [2,3].
> Following [3], our method can be combined with Earley’s algorithm [12], where our intermediate representations can replace dotted production rules.
> In our revision, we'll describe [2,3], and the idea of applying our method to [2,3].
>
> >It is unclear what new techniques or insights this work adds beyond existing grammar-based approaches for this task.
>
> We appreciate your pointing out the problem. We should've exactly distinguished previous work and our contributions.
> In our revision, we'll more exactly describe the difference between previous work and ours.
>
> >With only one dataset, it is hard to determine if the performance gains are dataset-specific or represent a robust advancement for grammar-based decoding.
>
> As you and the reviewer Ntkc commented, using only one dataset is a critical disadvantage of our paper.
> We acknowledge that experimenting on more than one dataset could enhance the validity of our method.
>
> However, at least in KQAPRO [1], we tried to compare our method with previous work as fairly as possible.
> We and previous work use the BART-base model, and we adapt hyperparameters from BART KoPL [1],
> Nevertheless, we admit that it cannot be an excuse for using only one dataset.
>
> **TODO**
>
> We're going to update our paper during the discussion period.
>
> - ReTraCk [13] will be added to "Related work".
> - Constrained decoding methods [2,3] for pretrained seq2seq will be added to "Related work".
>   - We will also briefly describe how our method can be combined with [2,3].
> - The difference between previous work and ours will be more clearly described.
>
> **Gratitude**
>
> We again greatly thank you for your reviews and feedback.
> If you have any more comments or questions, please feel free.

---

> ### Author Response · Authors · 2023-11-14
> **Response to Reviewer 4fJL (Part 2)**
>
> **Reference**
>
> [2] From Paraphrasing to Semantic Parsing: Unsupervised Semantic Parsing via Synchronous Semantic Decoding
> [3] Constrained Language Models Yield Few-Shot Semantic Parsers
> [5] A Syntactic Neural Model for General-Purpose Code Generation
> [6] TRANX: A Transition-based Neural Abstract Syntax Parser for Semantic Parsing and Code Generation
> [7] Neural Semantic Parsing with Type Constraints for Semi-Structured Tables
> [8] RAT-SQL: Relation-Aware Schema Encoding and Linking for Text-to-SQL Parsers
> [12] An Efficient Context-Free Parsing Algorithm
> [13] ReTraCk: A Flexible and Efficient Framework for Knowledge Base Question Answering
> [14] Compositional Generalization and Natural Language Variation: Can a Semantic Parsing Approach Handle Both?
> [15] UnifiedSKG: Unifying and Multi-Tasking Structured Knowledge Grounding with Text-to-Text Language Models
> [16] PICARD: Parsing Incrementally for Constrained Auto-Regressive Decoding from Language Models

---

> ### Comment · Reviewer_4fJL · 2023-11-21
> **Official Comment from Reviewer 4fJL**
>
> Hi Author,
>
> Thanks for your detailed response! While the author's response exhibits potential, it falls short of addressing the principal concerns. To merit a score increase, I recommend the author undertake at least one of the following actions:
>
> - **Zero-shot evaluation**: Although the author emphasizes the advantage of their approach in making language models compatible with pre-defined grammars without a specific decoder, this might not pose a significant issue for fine-tuned models based on my experience. However, it could be a substantial challenge for zero-shot setting. Providing additional evidence, even preliminary experimental results, particularly on zero-shot outcomes (perhaps based on gpt2 or other language models such as llama), would strengthen the credibility of the results.
>
> - **Add Benchmark Results**: It is essential for the author to extend the evaluation of their methods beyond the KQAPro benchmark. Exploring other benchmarks, such as other KBQA benchmarks or text-to-SQL benchmarks, would be acceptable. Additionally, given the combination of an autoregressive language model with predefined grammars under specific grammatical constraints, the author should acknowledge and differentiate their work from related works, such as UniSAR [1], especially if conducting experiments in the text-to-SQL domain. Clear articulation of distinctions, especially in comparison to UniSAR's constrained decoding, is crucial.
>
> - **Comparison to Previous Grammar-based Decoding Method Using the BART Encoder**: The author should conduct a comparative analysis of their method against previous grammar-based approaches that utilize a specific grammar-based decoder, restricting the usage to only the encoder part of the model. This comparison will provide a more comprehensive understanding of the proposed method's strengths and weaknesses in relation to existing techniques.
>
> Best,
>
> Reviewer 4fJL

---

> ### Author Response · Authors · 2023-11-23
> **Response to Reviewer 4fJL**
>
> We greatly thank you for the additional comment!
>
> If we perform some of the three additional experiments you suggested, our method would be more reliable.
>
> **About the suggested experiments**
>
> >Zero-shot evaluation
>
> We agree that learning a semantic parser with less supervision is important, and it would be good setting for our method.
> Thank you for the comment.
>
> >Add Benchmark Results
>
> As your previous comment, evaluating on more than one dataset would enhance validity of our method.
> In addition, we checked the paper of UniSAR, and we found that UniSAR also has similarity with our method in that both use trie data structures for constraints on structured data, such as KBs or DB schemas.
> Comparing with their method would be needed when experimenting in text-to-SQL domain.
> Thank you for providing us the paper.
>
> >Comparison to Previous Grammar-based Decoding Method Using the BART Encoder
>
> As your explanation, to emphasize the use of a pre-trained decoder, comparing with a custom decoder is reasonable.
> Thank you for making a good point.
>
> **Weakly-supervised setting**
>
> Unfortunately, we couldn't perform the suggested experiments.
> However, last night, we instead performed preliminary experiments for application to weakly-supervised learning (Appendix C).
> Since weakly-supervised learning is less addressed for pretrained seq2seq models, such as BART or T5, our results woudld be promising.
> We are going to develop the full process of weakly-supervised learning from now on.
>
> **Gratitude**
>
> We once again thank you so much for your feedback.

---

### Official Review · Reviewer_pMus · 2023-11-02

**Soundness:** 3 good
**Presentation:** 4 excellent
**Contribution:** 3 good
**Rating:** 6
**Confidence:** 4

**Summary:**

This paper proposes a grammar-based semantic parser for knowledge-based question answering. The grammars are specialized with fine-grained types and candidate expressions. During decoding, type constraints and candidate constraints can be enforced to search for the desired programs more effectively.

**Strengths:**

A well-executed work on designing grammars for knowledge-based question answering. The revisit of traditional grammar-based methods provides insights on whether prior information such as types are still useful in the current era of pre-trained models.  (But I also feel the question whether grammar-based constraints are still useful for large models needs to be studied more)

**Weaknesses:**

The main contribution, as the author points out,  is “To the best of our knowledge, our work is the first to use production rules as actions for semantic parsers based on pre-trained seq2seq models.”. First, I’m not sure what is the underlying challenge of extending traditional grammar-based seq2seqs to their pretrained counterparts? That is, I'm not sure about the technical contribution of the paper. Second, there are already existing works in this direction (as cited in the related work). For example, RAT-SQL (based on BERT) extends TRANX which expands production rules incrementally during decoding.

**Questions:**

- What is generation speed of the grammar-based parser compared with the baseline BART? I’m wondering that with more constraints, the parser might suffer from much slower inference speed
- would it be an issue if the size of the set of candidate expressions become too large? E.g., certain class (e.g., keyword-entity) has too many candidates.

---

> ### Author Response · Authors · 2023-11-14
> **Response to Reviewer pMus (Part 1)**
>
> We sincerely thank you for spending your precious time to review our paper.
>
> **Response to Questions**
>
> We first reply to your questions.
>
> >What is generation speed of the grammar-based parser compared with the baseline BART?
>
> Does "the baseline BART" mean BART KoPL [1]?
> We could run BART KoPL on the validation set by using the publicly released code [1].
> In the following table, we compare the average time to convert tokens of utterances to tokens of logical forms (or actions), where the batch size is 64.
>
> | Model                             | Time (ms) |
> |---|---|
> | BART KoPL                         |       4.5 |
> | Ours with $\Psi^\textrm{NONE}$    |       3.8 |
> | Ours with $\Psi^{\textrm{TYPE}-}$ |      10.0 |
> | Ours with $\Psi^\textrm{TYPE}$    |      10.0 |
> | Ours with $\Psi^\textrm{HYBR}$    |      10.2 |
>
> Our decoding with $\Psi^\textrm{NONE}$ is slightly faster than BART KoPL, although both don't use constraints during decoding.
> It's because the average length of action sequences of our model is shorter than the average length of output token sequences of BART KoPL; the average length is 28.8 for ours and 35.1 for BART KoPL.
>
> An example output sequence of BART KoPL is
> ```
> <s>FindAll <func> FilterStr <arg> official website <arg> http://www.thesiege.com/ <func> FilterConcept <arg> visual artwork <func> QueryAttrQualifier <arg> publication date <arg> 1999-01-21 <arg> place of publication</s>
> ```
> which is identical with the following logical form.
> ```
> (query-attr-qualifier (filter-concept (filter-str all-entities "official website" "http://www.thesiege.com/")
>                                       "visual artwork")
>                       "publication date" "1999-01-21" "place of publication")
> ```
>
> Since BART KoPL splits symbols (e.g. FindAll or FilterStr) into tokens (e.g. Find, All, Filter, Str), the model has slightly longer output sequences.
>
> >I’m wondering that with more constraints, the parser might suffer from much slower inference speed
>
> We think that additional contraints by types or candidate expressions don't much slow down the speed.
> Type constraints depend on valid actions $\Psi^\textrm{TYPE}(r({\boldsymbol{a}^\star}))$ for the current intermediate representation $r({\boldsymbol{a}^\star})$, and $\Psi^\textrm{TYPE}(r({\boldsymbol{a}^\star}))$ can be efficiently cached for the type of the leftmost non-terminal $\nu(r({\boldsymbol{a}^\star}))$.
> Constraints by candidate expressions depend on the time complexity of trie data structures.
> Retrieving valid actions $\Psi^\textrm{CAND}(r({\boldsymbol{a}^\star}))$ from a trie only takes $O(\text{length of prefix})$ time.
>
> >would it be an issue if the size of the set of candidate expressions become too large? E.g., certain class (e.g., keyword-entity) has too many candidates.
>
> The speed bottleneck of our constrained decoding is assigning $-\infty$ scores (Eq. 6) to all invalid actions ($\mathcal{A} - \Psi(r({\boldsymbol{a}^\star}))$), but this process can be efficiently computed in the most cases by GPUs.
> Initially, we make a mask tensor whose size is same with $|\mathcal{A}|$, which is the number of all actions, and update all the tensor's elements with $-\infty$ (processed by GPUs).
> Then, we can update the tensor depending on one of two cases:
> 1. if $|\Psi(r({\boldsymbol{a}^\star}))| < \frac{1}{2} |\mathcal{A}|$, we update the tensor's elements that correspond to $\Psi(r({\boldsymbol{a}^\star}))$ with 0 (processed CPUs).
> 2. if $|\Psi(r({\boldsymbol{a}^\star}))| > \frac{1}{2} |\mathcal{A}|$ (e.g. $\Psi(r({\boldsymbol{a}^\star}))$ = all production rules that generate nlt nodes), we update all tensor's elements with 0 (processed by GPUs) then update the tensor's elements that do not correspond to $\Psi(r({\boldsymbol{a}^\star}))$ (e.g. all production rules constructing compositional structures) with $-\infty$ (processed by CPUs).
> The computed tensor is added to log-probabilities of actions.
> Since $|\Psi(r({\boldsymbol{a}^\star}))|$ is usually very small or very large, the tensor is efficiently computed.
>
> In the worst case, if $|\Psi(r({\boldsymbol{a}^\star}))| = \frac{1}{2} |\mathcal{A}|$, GPUs cannot be efficiently exploited, then CPUs should take $O(\mathcal{A})$ time (c.f. $|\mathcal{A}|$ = 53 + 50,260).
> However, we think that the worst case does not easily occur.
> For your information, when given an empty prefix, the number of tokens that are returned from the trie $\tau(\texttt{keyword-entity})$ is 5,072; in addition, the number of tokens would be smaller when given a non-empty prefix.

---

> ### Author Response · Authors · 2023-11-14
> **Response to Reviewer pMus (Part 2)**
>
> **Additional Explanation**
>
> Please let us explain more about our paper with respect to your comments.
>
> >The main contribution, as the author points out, is “To the best of our knowledge, our work is the first to use production rules as actions for semantic parsers based on pre-trained seq2seq models.”.
> >First, I’m not sure what is the underlying challenge of extending traditional grammar-based seq2seqs to their pretrained counterparts? That is, I'm not sure about the technical contribution of the paper.
> >Second, there are already existing works in this direction (as cited in the related work). For example, RAT-SQL (based on BERT) extends TRANX which expands production rules incrementally during decoding.
>
> We regret using the arrogant expression "To the best of our knowledge ..." to describe the difference with previous neural seq2seq semantic parsers.
> At the time of writing, we hadn't seen semantic parsers that take production rules as actions and that are based on pretrained seq2seq models, such as BART or T5.
> Since we consider BERT, which are used by [8], as a pretrained auto-encoding model, which has no pretrained decoder, rather than a pretrained seq2seq model, we thought our model has difference with the previous semantic parsers.
> However, as you commented, applying [5,6,7] to BART or T5 is not technically difficult.
> In addition, we've recently found other previous work [2,3] that take tokens (not production rules) as actions but constrain the tokens by using production rules.
> We'll remove "To the best of our knowledge ..." and add descriptions of [2,3].
>
> **TODO**
>
> We're going to update our paper during the discussion period.
>
> - The sentence beginning with "To the best of our knowledge ..." will be removed.
> - Constrained decoding methods [2,3] for pretrained seq2seq will be added to "Related work".
>   - We will also briefly describe how our method can be combined with [2,3].
>
> **Gratitude**
>
> We again greatly thank you for your reviews and feedback.
> If you have any more comments or questions, please feel free.
>
> **Reference**
>
> [1] KQA Pro: A Dataset with Explicit Compositional Programs for Complex Question Answering over Knowledge Base
> [2] From Paraphrasing to Semantic Parsing: Unsupervised Semantic Parsing via Synchronous Semantic Decoding
> [3] Constrained Language Models Yield Few-Shot Semantic Parsers
> [5] A Syntactic Neural Model for General-Purpose Code Generation
> [6] TRANX: A Transition-based Neural Abstract Syntax Parser for Semantic Parsing and Code Generation
> [7] Neural Semantic Parsing with Type Constraints for Semi-Structured Tables
> [8] RAT-SQL: Relation-Aware Schema Encoding and Linking for Text-to-SQL Parsers

---

> ### Comment · Reviewer_pMus · 2023-11-16
> **Clarification of technical novelty**
>
> Thanks for the response!
>
> Though the authors acknowledge more related work, it's still unclear what is the main technical contribution. As I read from the response to other reviewers, the authors seem to suggest that the main challenge of combing grammar +  pretrained models is to handle the vocabulary discrepancy (i.e., pretrained models have their own tokenizers which is not the same vocabulary used in grammar), which I think is not a significant issue. Regarding the future work mentioned on combining Earley with pretrained decoder, there are very related work in this direction[1,2].  And I think the way of handling candidate expression can be viewed as a special case of Earley-based constraining used in [1,2].
>
> [1] Synchromesh: Reliable code generation from pre-trained language models
>
> [2] Grammar Prompting for Domain-Specific Language Generation with Large Language Models

---

> ### Author Response · Authors · 2023-11-17
> **Clarifying technical novelty**
>
> We greatly thank you for the additional comment!
>
> The two latest papers [17,18] you inform us about are related to our work and our vague idea for future work.
> We reorganize our thoughts with respect to
> the scope of work, contributions, and future work.
> We very appreciate your feedback!
>
> **Scope of Work**
>
> We deal with the problem of developing a semantic parser that is fine-tuned from a pre-trained seq2seq model.
> Especially, the semantic parser constructs a logical form that contains components (e.g., entities, relations) from a KB which could be large.
> We assume the KB components in logical forms (or intermediate representations) are represented in naturual language.
>
> **Contributions**
>
> Our main contribution is combining the two previous constrained decoding methods for
> - constructing compositional structures [5,7] and
> - generating valid KB components, which are represented as candidate expressions [4]
>
> The resulting method is designed with respect to the following principles:
> - Scalability
>   - $\Psi^\textrm{TYPE}$ and $\Psi^\textrm{CAND}$ can be scaled up for various types and many KB components
>   - Related Work: Synchromesh [17]
>     - [17] also uses context-sensitive constraints on structured data (e.g., constraints on DB schemas)
>     - However, their constrained decoding is inappropriate to exploit many elements of structured data (e.g., entities of KBs)
>     - [17] dynamically constructs a regular expression, which represents all possible next tokens, for context-sensitive constraints
>     - Therefore, defining a constraint on many elements of structured data is costly and results in a huge regular expression
> - Efficiency
>   - $\Psi^\textrm{HYBR}$ instantly retrieves all valid actions $\Psi^\textrm{HYBR}(r({\boldsymbol{a}^\star}))$ for a given $r({\boldsymbol{a}^\star})$.
>   - We also generalize the [PrefixConstrainedLogitsProcessor](https://github.com/huggingface/transformers/blob/913d03dc5e78b82c24be7a52c9ad06dd1022f1e2/src/transformers/generation/logits_process.py#L1052) method [4] to work with our semantic parser. It corresponds to the method that computes a mask tensor, which is the speed bottleneck, described in the first reply to pMus.
>   - Related work: PICARD [16], Synchromesh [17]
>     - [16,17] devised a predicate which takes a prefix of a logical form then returns a boolean value whether the prefix is valid for a logical form language
>     - [16] only consider tokens with top-k probabilities
>     - [17] test many tokens, although they use trie data structures to keep "rejected" tokens and skip testing for the tokens that have the rejected tokens as prefixes.
>     - [22] also points out high overhead of existing constrained decoding for seq2seq models
> - Effectiveness
>   - The ablation study shows that performance benefits from candidate expressions
>   - A node class with more candidate expressions usually contributes more to the performance
>   - Therefore, constraints on many elements of KBs are important
> - Generalization
>   - Our method can be applied to many types of KB component categories
>   - Our method is generalization of the constrained decoding by [19]
>
> **Future work**
> - Weakly-supervised learning [7,20,21]
>   - An initial semantic parser, which is less trained, can efficiently make various search branches, since all valid actions $\Psi^\textrm{HYBR}(r({\boldsymbol{a}^\star}))$ are instantly retrieved.
>
> **TODO**
>
> We're going to update our paper to include the content of this comment.
>
> **Gratitude**
>
> We once again thank you so much for your feedback.
>
> **Reference**
>
> [2] From Paraphrasing to Semantic Parsing: Unsupervised Semantic Parsing via Synchronous Semantic Decoding
> [3] Constrained Language Models Yield Few-Shot Semantic Parsers
> [4] Autoregressive Entity Retrieval
> [5] A Syntactic Neural Model for General-Purpose Code Generation
> [7] Neural Semantic Parsing with Type Constraints for Semi-Structured Tables
> [16] PICARD: Parsing Incrementally for Constrained Auto-Regressive Decoding from Language Models
> [17] Synchromesh: Reliable Code Generation from Pre-trained Language Models
> [18] Grammar Prompting for Domain-Specific Language Generation with Large Language Models
> [19] TIARA: Multi-grained Retrieval for Robust Question Answering over Large Knowledge Base
> [20] Learning Dependency-Based Compositional Semantics
> [21] Iterative Search for Weakly Supervised Semantic Parsing
> [22] Unveiling the Black Box of PLMs with Semantic Anchors: Towards Interpretable Neural Semantic Parsing

---

### Official Review · Reviewer_Ntkc · 2023-11-03

**Soundness:** 4 excellent
**Presentation:** 4 excellent
**Contribution:** 2 fair
**Rating:** 5
**Confidence:** 4

**Summary:**

This paper introduces a grammar augmented with candidate expressions for KB-QA. The grammar decoder enforces the type of candidate expressions during the decoding phrase. The proposed method achieves the state-of-the-art performance on KQAPRO datasets. The ablation study shows the effectiveness of the method. However, the paper fails to show the proposed method works for other KB-QA tasks. And it is not sure what is the scope of the problem the proposed method could outperform the beselines.

**Strengths:**

* The method is proposed for KBQA tasks is noval.
* A state-of-the-art performance on KQAPRO
* The abalation study shows the effectiveness of this method

**Weaknesses:**

* In this paper, the performance of the model is recorded only in one KB-QA dataset. It is not known how good is this method among all KB-QA tasks.

**Questions:**

* Are there any results on KB-QA datasets other than KQAPRO for the proposed method? It would be better to show the performance of the proposed method on more than one dataset. The datasets could be (but not limited to) Complex Web Questions and Grail QA.

* [Not critical] This paper compares their decoding method with Picard's paper to show the proposed method has a good latency. Why are the two models in different tasks (KBQA vs Spider), different model architectures (BART vs T5), and different hardwares comparable?

---

> ### Author Response · Authors · 2023-11-14
> **Response to Reviewer Ntkc**
>
> We sincerely thank you for spending your precious time to review our paper.
>
> **Response to Questions**
>
> We first reply to your questions.
>
> >Are there any results on KB-QA datasets other than KQAPRO for the proposed method? It would be better to show the performance of the proposed method on more than one dataset. The datasets could be (but not limited to) Complex Web Questions and Grail QA.
>
> We haven't performed experiments on another dataset, although we're considering other datasets, whose tasks could be KBQA, Text-to-SQL or others.
> As you and the reviewer 4fJL commented, using only one dataset is a critical disadvantage of our paper.
> We acknowledge that experimenting on more than one dataset could enhance the validity of our method.
>
> However, at least in KQAPRO [1], we tried to compare our method with previous work as fairly as possible.
> We and previous work use the BART-base model, and we adapt hyperparameters from BART KoPL [1],
> Nevertheless, we admit that it cannot be an excuse for using only one dataset.
>
> >[Not critical] This paper compares their decoding method with Picard's paper to show the proposed method has a good latency. Why are the two models in different tasks (KBQA vs Spider), different model architectures (BART vs T5), and different hardwares comparable?
>
> We wanted to compare our method with constrained decoding for __pretrained seq2seq model__, such as BART or T5.
> At the time of writing, PICARD was only method we know for the scenario (however, we've recently found other previous work [2,3]).
> We think that methods based on production rules have better efficiency, so we wanted to emphasize it.
> As your comment, the comparison would be fair if we could implement our method on both Spider and KQA Pro.
>
> With respect to hardware, CPUs and GPUs may affect latency, so we specified the exact spec of our own hardware, where our model could run __solely__ in the machine.
> However, when we measured latency on other environments (e.g. a cluster with V100), the latency was not much different.
>
> **Additional Explanation**
>
> Please let us explain more about our paper with respect to your comments.
>
> >And it is not sure what is the scope of the problem the proposed method could outperform the beselines.
>
> We appreciate your pointing out the problem. As your comment, the scope, where our method can be effectively applied, is less explained in our paper.
> Semantic parsing on KBs (or structured data) produces logical forms that include KB components (e.g. entities or relations).
> However, if some KB components are hardly seen or unseen during training, semantic parser cannot properly infer the correct expressions of the KB components, especially when the counterparts in NL utterances have different expressions.
> For example, a KB component (relation) is "country of citizenship" and its NL counterpart is "a citizen of" (Figure 1).
> This problem usually occurs when a KB is large (e.g. KQAPRO), or when a KB is updated after training (e.g. adding new entities) [4].
> We will add more explanation about the scope of the problem.
>
> **TODO**
>
> We're going to update our paper during the discussion period.
>
> - Constrained decoding methods [2,3] for pretrained seq2seq will be added to "Related work".
>   - We will also briefly describe how our method can be combined with [2,3].
> - Explanation about "the scope of the problem" where our method can be effectively applied will be added to one of "Introduction", "Related Work" or "Appendix".
>
> **Gratitude**
>
> We again greatly thank you for your reviews and feedback.
> If you have any more comments or questions, please feel free.
>
> **Reference**
>
> [1] KQA Pro: A Dataset with Explicit Compositional Programs for Complex Question Answering over Knowledge Base
> [2] From Paraphrasing to Semantic Parsing: Unsupervised Semantic Parsing via Synchronous Semantic Decoding
> [3] Constrained Language Models Yield Few-Shot Semantic Parsers
> [4] Autoregressive Entity Retrieval

---

### Author Response · Authors · 2023-11-23
**Summary by Authors**

We sincerely thank the reivewers for all the helpful comments.
It was a precious opportunity for us to evaluate our paper objectively and to improve the paper with your advice.
We greatly appreciate for your time for reading our paper and giving us great comments.

**Summary**

As the final comment, we summarize our paper and updated contents.

**Contributions**

- We propose a grammar augmented with candidate expressions for semantic parsing on __large__ KBs with a sequence-to-sequence (seq2seq) pre-trained language model (PLM)
  - seq2seq PLMs have both pre-trained encoders and decoders (e.g., BART, T5)
- Unlike previous grammar for semantic parsing on seq2seq PLMs or large language models (LLMs), our grammar incorporates large information of KBs
  - The information consists of KB components, such as entities or relations, and categories that the components belong to
- Our grammar is scalable and efficient to address various types [5] and many candidate expressions [4]
  - Contraints by types and candidate expressions are seamlessly unified on intermediate representations
  - We developed an efficient constrained decoding algorithms that generalize the method used in [4] (Appendix B)
- Our semantic parser achieved the state-of-the-art on KQAPRO [1]
- Our grammar is promising for weakly-supervised learning as preliminary experiments show good results (Appendix C)
  - Since weakly-supervised semantic parsing with seq2seq PLMs is less addressed in previous work, our work could be important

**Update**

Major update
- Introduction
  - Emphasizing our main contribution: constraints on a large KB for seq2seq PLMs
  - Clarifying the scope of the problem; semantic parsing with seq2seq PLMs
- Related Work
  - We cited additional papers including the papers provided by reviewers [2,3,13,17,18, ...]
- Appendix
  - Appendix B: Algorithms for decoding speed optimization
  - Appendix C: Preliminary experiments for weakly-supervised learning
  - Appendix D: Idea about Earley's algorithm, which can be applied to both seq2seq PLMs and LLMs

Minor update
- The decoding speed about PICARD is removed, since its domain is different with ours
- Notation addition
  - $\mathcal{A^\textrm{COM}}$: All actions for constructing compositional structures or atomic units (Section 2)
  - $\mathcal{A^\textrm{NLT}}$: All actions that generate nlt nodes (Section 2)

**TODO**

We are going to develop the full process of weakly-supervised learning with our grammar.
If we have the opportunity for the final revision, we would add more results on weakly-supervised learning.

**Gratitude**

We greatly thank reviewers for the comments and the discussions!

**Reference**

[1] KQA Pro: A Dataset with Explicit Compositional Programs for Complex Question Answering over Knowledge Base
[2] From Paraphrasing to Semantic Parsing: Unsupervised Semantic Parsing via Synchronous Semantic Decoding
[3] Constrained Language Models Yield Few-Shot Semantic Parsers
[4] Autoregressive Entity Retrieval
[5] A Syntactic Neural Model for General-Purpose Code Generation
[6] TRANX: A Transition-based Neural Abstract Syntax Parser for Semantic Parsing and Code Generation
[7] Neural Semantic Parsing with Type Constraints for Semi-Structured Tables
[8] RAT-SQL: Relation-Aware Schema Encoding and Linking for Text-to-SQL Parsers
[9] Building a Semantic Parser Overnight
[10] On The Ingredients of an Effective Zero-shot Semantic Parser
[11] Weakly Supervised Semantic Parsing by Learning from Mistakes
[12] An Efficient Context-Free Parsing Algorithm
[13] ReTraCk: A Flexible and Efficient Framework for Knowledge Base Question Answering
[14] Compositional Generalization and Natural Language Variation: Can a Semantic Parsing Approach Handle Both?
[15] UnifiedSKG: Unifying and Multi-Tasking Structured Knowledge Grounding with Text-to-Text Language Models
[16] PICARD: Parsing Incrementally for Constrained Auto-Regressive Decoding from Language Models
[17] Synchromesh: Reliable Code Generation from Pre-trained Language Models
[18] Grammar Prompting for Domain-Specific Language Generation with Large Language Models
[19] TIARA: Multi-grained Retrieval for Robust Question Answering over Large Knowledge Base
[20] Learning Dependency-Based Compositional Semantics
[21] Iterative Search for Weakly Supervised Semantic Parsing
[22] Unveiling the Black Box of PLMs with Semantic Anchors: Towards Interpretable Neural Semantic Parsing

---

### Public Comment · ~Ada_Wan1 · 2023-11-23
**Please sort out what role "meaning" plays in the context of statistical computing first.**

1. This paper suggests that semantic parsing is necessary in order to extract information for knowledge base (KB) question answering, but did not reconcile with the statistics and surface strings of relevant elements of either the KBs or the pre-trained language model. (As we know, also from e.g. [1], that language models are statistics-driven.)

2. Re "Since logical forms contain representations of KB components, the information of KBs is necessary for semantic parsers to generate valid logical forms. Therefore, incorporating the large information of KBs into grammars is important, but scalable and efficient designs of the grammars are inevitable for practical use.":
it seems like there is some confusion in these statements, or the motivation in general. Why is the intermediate structure/representation of grammar important, when one can simply work with the information of the KBs and the model?

3. What are the authors' justification for how "meaning" is to matter in the context of information processing and statistical computing?

4. Re "However, the seq2seq models have difficulty in learning to generate logical forms that contain components drawn from large KBs.":
what are the reasons to expect the seq2seq models to generate something that is out of vocabulary used in training? Is there prior work that compares vocabulary of input and output data?

Please note:
a sufficient and more rigorous statistical evaluation of your work (i.e. beyond the tables in the paper) is not an option, it is a basic requirement, a must. Even if one were to aim at obtaining "meaningful sequences" (however that should be qualified) through information processing, one must have the valid computational, statistical requirements clarified/satisfied. This goes for all "symbolic processing/interpretation" --- e.g. are there statistical correlations between meaning and information? To what extent is "meaning" relevant in computing? These are questions that should have been addressed first.

[1] Ada Wan. Fairness in representation for multilingual NLP: Insights from controlled experiments on conditional language modeling. In International Conference on Learning Representations, 2022. https://openreview.net/forum?id=-llS6TiOew.

---

### Meta-Review · Area_Chair_cSVn · 2023-12-05

**Metareview:**

The paper has some potential, but many issues with the contribution being relatively thin compared to previous work and the narrowness of benchmarking. The reviewers provided concrete suggestions for the authors to follow, and these seems appropriate and necessary to bring this work above the bar for acceptance.

**Justification For Why Not Higher Score:**

Thinness of contribution and experiments.

**Justification For Why Not Lower Score:**

N/A

---

### Decision · Program_Chairs · 2024-01-16

Reject